# Comprehensive transcriptome analysis of cochlear spiral ganglion neurons at multiple ages

Chao Li[1†], Xiang Li[1†], Zhenghong Bi[1†], Ken Sugino[2], Guangqin Wang[1], Tong Zhu[1], Zhiyong Liu[1,3]*

[1]Institute of Neuroscience, State Key Laboratory of Neuroscience, CAS Center for Excellence in Brain Science and Intelligence Technology, Chinese Academy of Sciences, Shanghai, China; [2]Howard Hughes Medical Institute, Janelia Research Campus, Ashburn, United States; [3]Shanghai Center for Brain Science and Brain-Inspired Intelligence Technology, Shanghai, China

**Abstract** Inner ear cochlear spiral ganglion neurons (SGNs) transmit sound information to the brainstem. Recent single cell RNA-Seq studies have revealed heterogeneities within SGNs. Nonetheless, much remains unknown about the transcriptome of SGNs, especially which genes are specifically expressed in SGNs. To address these questions, we needed a deeper and broader gene coverage than that in previous studies. We performed bulk RNA-Seq on mouse SGNs at five ages, and on two reference cell types (hair cells and glia). Their transcriptome comparison identified genes previously unknown to be specifically expressed in SGNs. To validate our dataset and provide useful genetic tools for this research field, we generated two knockin mouse strains: *Scrt2-P2A-tdTomato* and *Celf4-3xHA-P2A-iCreER-T2A-EGFP*. Our comprehensive analysis confirmed the SGN-selective expression of the candidate genes, testifying to the quality of our transcriptome data. These two mouse strains can be used to temporally label SGNs or to sort them.

*For correspondence:
zhiyongliu@ion.ac.cn

[†]These authors contributed equally to this work

**Competing interests:** The authors declare that no competing interests exist.

## Introduction

Genome-wide profiling analyses such as RNA-Seq are being increasingly used for dissecting the mechanisms underlying cell-fate determination, development, and disease progression. RNA-Seq can be employed to address several developmental questions, including these two key questions: (1) Which genes are expressed specifically in one cell type but not others within one tissue and/or organ? (2) Which genes exhibit expression patterns that change dynamically at distinct developmental ages within the same cell type? RNA-Seq analyses fall within two main categories: the first is Bulk-Seq, where the data represent an average of gene-expression patterns from multiple cells (*Ozsolak and Milos, 2011*), and although this allows for vast sequencing coverage (or depth), the incurred cost is that variations among the pooled cells are obscured. The second is single-cell RNA-Seq, which enables assessment of gene expression at unprecedented single-cell resolution (*Tang et al., 2013*), but the coverage and detection sensitivity here are typically lower than those in Bulk-Seq. Because the two approaches complement each other in their strengths, the optimal strategy would be to run both Bulk-Seq and single-cell RNA-Seq, thus obtaining both sequencing depth and single-cell-resolution data. Notably, this combined approach was used in two recent studies examining mouse inner ear auditory and utricle cells (*Jen et al., 2019*; *Yamashita et al., 2018*).

In mouse cochlea, two types of hair cells (HCs) coexist: outer hair cells (OHCs) and inner hair cells (IHCs) and they differ in function and innervation (*Kelley, 2006*). Whereas OHCs act as sound amplifiers and are innervated by Type II cochlear spiral ganglion neurons (SGNs), IHCs form synapses with Type I SGNs. Type I SGNs can be divided into three subtypes, Ia, Ib, and Ic, as revealed by three

recent single-cell RNA-Seq studies of mouse SGNs (*Petitpré et al., 2018*; *Shrestha et al., 2018*; *Sun et al., 2018*). Multiple genes, such as *Calb2*, *Pou4f1*, and *Lypd1*, show markedly different expression patterns across the three subtypes. However, these studies focused on transcriptomic differences among distinct SGN subtypes: no comparisons were performed between the transcriptomes of SGNs and non-SGN cells (i.e. HCs or glia). Thus, the identity of the genes that are specifically expressed in SGNs but not in other cell types within the inner ear remains unclear, as does the temporal dynamics of these genes during SGN differentiation (i.e. it is not known which genes show notable expression differences at distinct developmental stages). However, cell-type specificity and temporal specificity are both critical features that should be presented by genes that regulate SGN fate determination and maturation. Therefore, identification of the genes that possess these properties represents a critical first step that will enable the function of these genes to be studied. In addition to advancing our understanding of SGN development, identifying these key regulators will also likely facilitate studies focusing on regenerating degenerated or damaged SGNs from other cells such as cochlear glial cells (*Noda et al., 2018*).

Here, we addressed the aforementioned two questions, on cell-type specificity and temporal specificity of SGN genes, by employing bulk RNA-Seq covering mRNAs and other types of RNA (*Liu et al., 2015*). We manually picked SGNs at five distinct ages: embryonic day 15.5 (E15.5), postnatal day 1 (P1), P8, P14, and P30. To identify SGN-specific genes, we compared the transcriptomes of SGNs with those of inner ear HCs at P12 and cochlear glial cells at P8, which resulted in the discovery of multiple previously unknown SGN-specific genes. These SGN-specific genes were further divided into genes exhibiting constant or dynamic expression. Moreover, we analyzed the dynamics of genes expressed in SGNs, regardless of whether they are expressed in HCs and glia. To further validate our transcriptome analysis, 8 SGN-specific genes were analyzed using RNA in situ hybridization, and from these genes, *Scrt2* and *Celf4* were selected for generating, respectively, *Scrt2-P2A-tdTomato* and *Celf4-3xHA-P2A-iCreER-T2A-EGFP* knockin mice. Characterization of these two mouse strains showed that *Scrt2* and *Celf4* are expressed in SGNs and exhibit constant expression (from early embryonic to adult ages) and dynamic expression, respectively. Thus, this work provides a comprehensive transcriptome analysis, with high-quality data and deep sequencing coverage, of SGNs at five developmental ages. Furthermore, the two new mouse strains developed here will help future studies aimed at sorting SGNs or temporally manipulating gene expression in SGNs. Lastly, because *Scrt2* and *Celf4* are also expressed in the central nervous system, the two knockin mouse lines should also serve as useful tools for brain research in general.

## Results

### Isolated and purified SGNs are highly enriched in neuronal genes and depleted in HC and glial genes

Cochlear SGNs transiently and specifically express *Sonic Hedgehog* (*Shh)* (*Bok et al., 2013*; *Liu et al., 2010*; *Tateya et al., 2013*), and we used this gene to isolate SGNs. Briefly, our previous fate-mapping study showed that *ShhCre/+* exclusively labels SGNs in the cochlea (*Liu et al., 2010*). In cochlear tissues of *ShhCre/+; Rosa26-CAG-loxp-stop-loxp-tdTomato* (Ai9)/+ mice, tdTomato+ cells were observed in the ganglion area at P1 (*Figure 1A–A'*). These tdTomato+ cells expressed the neuronal marker Mafb (*Yu et al., 2013*), but not glial marker Sox10 (*Li et al., 2018*) (*Figure 1B–B'''*), confirming that the tdTomato+ cells were SGNs and not their neighboring glial cells. In this study, tdTomato+ SGNs were obtained by our routinely employed manual picking/washing approach (*Hempel et al., 2007*; *Li et al., 2018*; *Liu et al., 2015*). To minimize gene expression profiling alteration in the in vitro condition, we used a total of 3 ~ 3.5 hr from mice euthanasia to cell preservation in lysis buffer.

SGNs were picked at five different ages: E15.5, P1, P8, P14 and P30. E15.5 was the earliest age at which we could visualize a bright tdTomato signal and pick tdTomato+ SGNs confidently (*Figure 1C*). Three replicates were performed per age, and each replicate contained ~100 SGNs. We hypothesized that these five ages should capture the main signatures of gene expression profiles during the process of SGN differentiation. SGNs are likely highly immature at E15.5 and P1, actively differentiating at P8, well differentiated at P14 as mice start detecting sound, and fully differentiated and functional at P30. Besides picking SGNs, we isolated, as reference cells, HCs (vestibular and

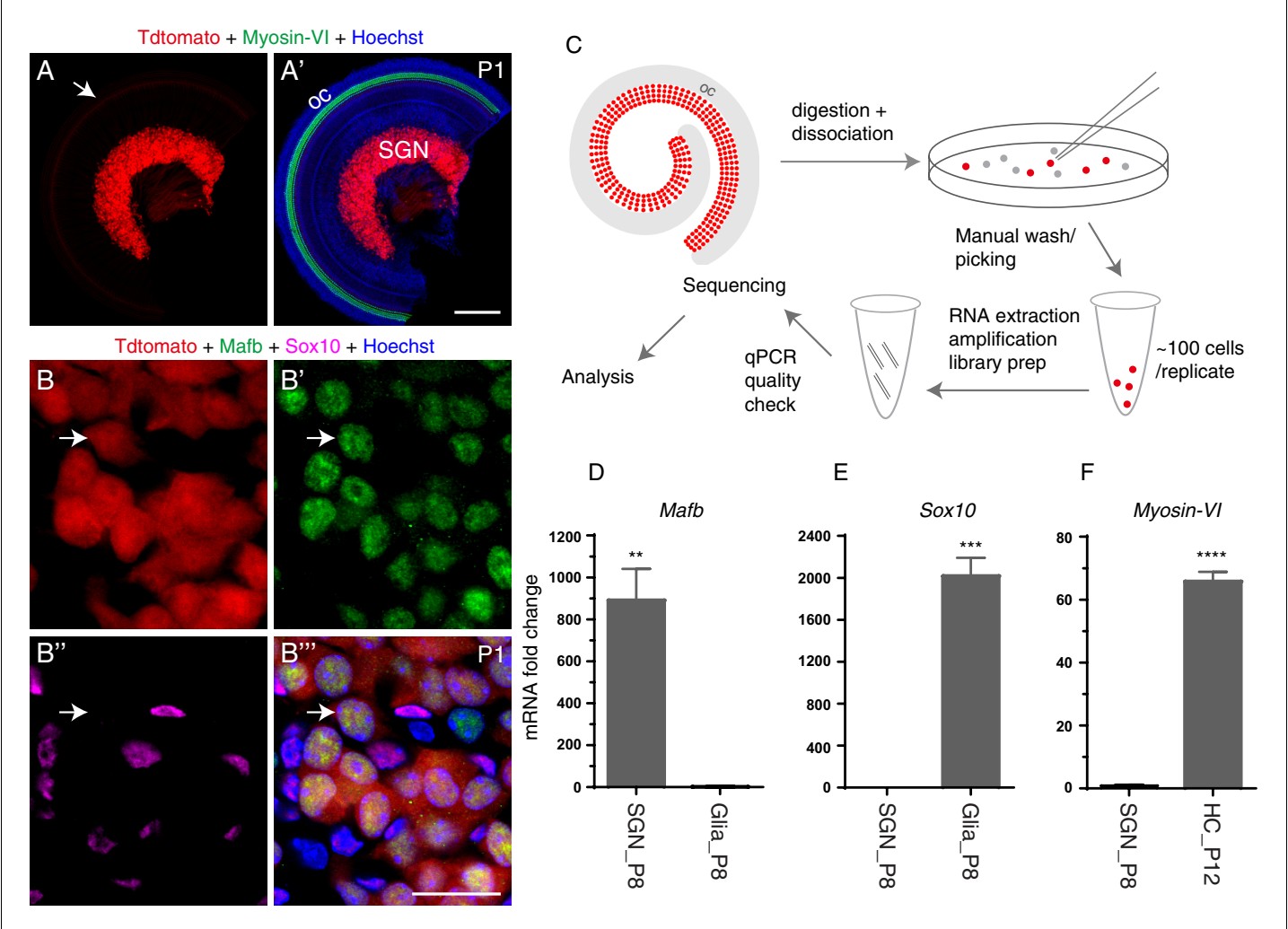

**Figure 1.** Genetic model, bulk RNA-Seq methodology, and qPCR quality check. (A–A') SGNs were endogenously labeled with tdTomato at P1 (or other ages after E15.5) in the mouse line *ShhCre/+; Rosa26-CAG-loxp-stop-loxp-tdTomato/+*. Arrow (in A) indicates SGN fibers with weak tdTomato expression innervating Myosin-VI+ HCs (green in A') in the organ of Corti (oc). (B–B''') Cochlear tissues triple-labeled with Mafb and Sox10 antibodies and endogenous tdTomato fluorescence. Arrows: one tdTomato+ SGN expressing the neuronal marker Mafb but not the glial marker Sox10. (C) Illustration of our experimental procedures. Red SGNs were manually picked under a fluorescence microscope, washed three times, and placed in lysis buffer. (D–F) qPCR analysis of three genes, *Mafb* (D), *Sox10* (E), and *Myosin-VI* (F). SGNs passed our quality check if they showed significant enrichment of *Mafb* but depletion of *Sox10* and *Myosin-VI*; otherwise, the SGNs were discarded. **$p<0.01$, ***$p<0.001$, ****$p<0.0001$. Scale bars: 200 $\mu$m (A') and 20 $\mu$m (B''').

The online version of this article includes the following source data and figure supplement(s) for figure 1:

**Source data 1.** Neuronal gene was enriched, HC and glial genes were depleted in SGNs at P8.

**Figure supplement 1.** qPCR analysis of *Mafb*, *Sox10*, and *Myosin-VI* in SGNs at E15.5, P1, P14, and P30.

**Figure supplement 1—source data 1.** Neuronal gene was enriched, HC and glial genes were depleted in SGNs at E15.5, P1, P14 and P30.

cochlear) at P12 and cochlear glial cells at P8. The methods section contains detailed information regarding the HCs and glial cells. Briefly, gene-expression profiles of HCs and glia were used as a reference database for quality-check analyses and for revealing SGN-specific genes (showing constant or dynamic expression) within the inner ear, which was our primary goal in this study.

Before fragmenting full-length cDNA derived from SGNs at the different ages, qPCR analysis was performed to determine the purity of the collected cells. For a sample to pass this first quality check point (pre-sequencing stage), the sample had to be significantly enriched in the neuronal gene *Mafb* and depleted in both the glial gene *Sox10*, and HC gene Myosin-VI (*Myo6*) (*Avraham et al., 1995*). For example, in SGNs at P8, *Mafb* expression was enriched 890.7 ± 150.2 fold, as compared to glia

cells (n = 3 sample replicates) (*Figure 1D*), whereas, *Sox10* expression was reduced by 2016.7 ± 174.1 fold (n = 3 sample replicates) (*Figure 1E*). Similarly, *Myosin-VI* expression was 65.7 ± 3.1 fold lower in SGNs than in HCs (n = 3 sample replicates) (*Figure 1F*). Similar qPCR quality checks were performed for SGNs at other four ages (*Figure 1—figure supplement 1*). Collectively, our data suggest that the isolated SGN samples were pure and that the cDNA was of high quality. Only the samples that passed this qPCR quality control were proceeded into library construction and paired-end sequencing.

## Acquiring high quality transcriptome data

On average, there were 21.5 million sequence reads per library, of which ~ 85% were mapped to the mm10 genome (*Figure 2—figure supplement 1A–B*). Per sample, approximately 15,000 genes were detected with a TPM (transcripts per million) threshold of 1 (*Figure 2—figure supplement 1C*), and ~6000 of these genes featuring TPM values of >10 (*Figure 2—figure supplement 1D*). Pearson's correlation coefficient analysis revealed that the general transcriptomes of SGNs were highly distinct from those of HCs and glial cells (*Figure 2—figure supplement 1E*). Moreover, SGNs of different ages clustered into distinct groups, which supported the hypothesis that global gene profiles dynamically change throughout SGN differentiation.

Before searching for genes showing constant or dynamic expression, we performed a second quality check at this post-sequencing stage to further ensure the purity of the SGN samples. We used additional gene markers that together covered 10 different cell types (*Figure 2—figure supplement 2*). In addition, we also examined three house-keeping genes, *Gapdh*, *Actg1* and *Actb*, all of which were detected in all samples (*Figure 2—figure supplement 2*). Again, we found that SGNs were significantly enriched with neural genes *Mafb*, *Slc17a6*, *Slc17a7* and *Lypd1*, but dramatically depleted of genes of HC-specific (*Myo6*, *Myo7a*, *Pou4f3* and *Gfi1*) and glial-specific (*Sox10*, *Foxd3* and *Plp1*) (*Figure 2—figure supplement 2*). We included both vestibular and cochlear HCs to cover most of the genes expressed in inner ear HCs. Vestibular HCs are known to express *Sox2*, which was accordingly detected here in the HC populations (*Figure 2—figure supplement 2*). Together, the quality checks at the pre-sequencing stage (*Figure 1D–F* and *Figure 1—figure supplement 1*) and post-sequencing stage (*Figure 2—figure supplement 2*) ensured that the SGN samples were highly pure and that the RNA-Seq data were of high quality.

## Identifying SGN-specific genes with constant and dynamic expression pattern

We first aimed to identify genes that are both highly and specifically expressed in SGNs but are not expressed in HCs or glia. For example, *Parvalbumin* and *Calbindin* are expressed in both HCs and SGNs, and are not our target genes (*Liu et al., 2012*). We classified the SGN-specific genes into two categories based on their expression during development: constant and dynamic. Computational analysis revealed 21 SGN-specific genes showing constant expression patterns in SGNs at all time-points examined (*Figure 2A*), and 68 SGN-specific genes showing dynamic expression patterns (*Figure 2B*). Notably, *Mafb*, which showed dynamic expression and was included in *Figure 2B*, is known to decrease between E16.5 and P15 in SGNs and to be critical for SGN development (*Yu et al., 2013*). This suggests that although our bioinformatics analysis for defining SGN-specific genes was highly stringent, it was still sensitive. Because most of the identified genes were not previously known to be SGN-specific, we validated the following eight genes (four from each category) by using RNA in situ hybridization at P1: *Elavl4*, *Esrrg*, *Scrt1*, *Scrt2*, *Stmn3*, *Shox2*, *Gabbr2*, and *Celf4*. All eight genes were specifically expressed in SGN region (*Figure 2—figure supplement 3*), which again validated the quality of our RNA-Seq data. Note that genes above were SGN-specific within the cochlea organ. As to be discussed below, *Scrt2* and *Celf4* were also expressed in vestibular neurons (VGNs) and neurons in the central nervous system.

## Classifying dynamics of genes expressed in SGNs

Next, we analyzed genes that were expressed in SGNs, regardless of their patterns in HCs and glias, because genes shared by these cell types may also regulate SGN development (*Figure 3*). For this analysis, one of three states, up-regulated (u), down-regulated (d) and unchanged (-) was assigned according to whether a gene was significantly (q < 0.05) up-regulated (fold change (FC) >2) and

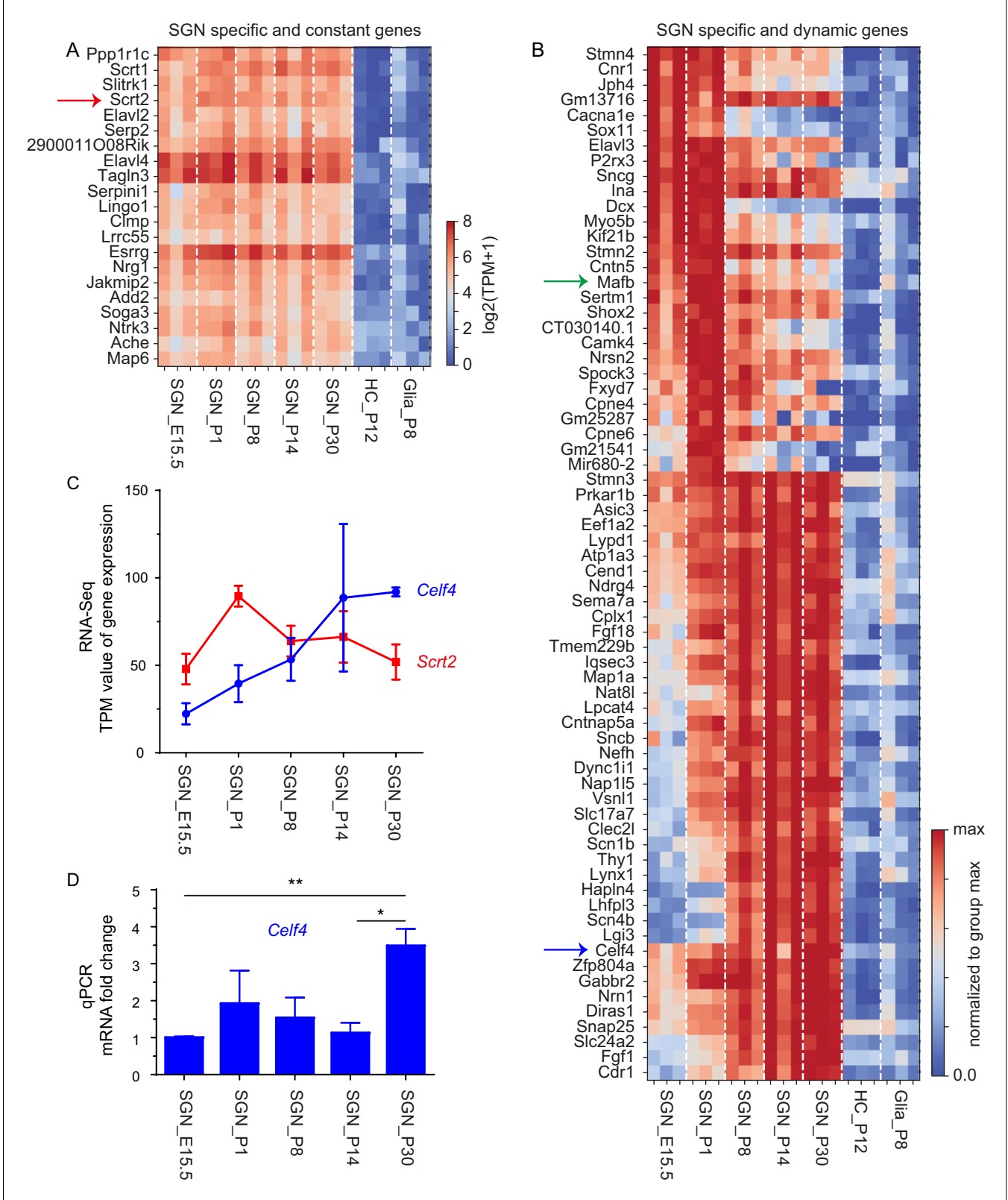

**Figure 2.** SGN-specific genes showing constant and dynamic expression. (**A**) Our selection criteria for constant expression were met by 21 SGN-specific genes, which were expressed in SGNs at all ages but not expressed in HCs or glia. (**B**) Dynamic expression was exhibited by 68 SGN-specific genes; the expression levels of these genes in SGNs changed significantly across ages, being either lower at younger ages but higher at older ages, or vice versa. For these genes to be considered bona fide SGN-specific genes, they had to be depleted in HCs and glial cells. (**C**) *Scrt2* and *Celf4* are used here as *Figure 2 continued on next page*

*Figure 2 continued*

examples to illustrate expression levels across different ages. Y-axis: expression in transcripts per million (TPM); data are shown as means ± SEM. *Scrt2* (red): SGN-specific gene showing constant expression; *Celf4* (blue): SGN-specific gene showing dynamic expression. (D) qPCR analysis of *Celf4* in SGNs at all five ages. *Celf4* mRNA levels were significantly different between E15.5 and P30 (**$p<0.01$) and between P14 and P30 (*$p<0.05$).

The online version of this article includes the following source data and figure supplement(s) for figure 2:

**Source data 1.** Scrt2 and Celf4 expression level by RNA-Seq and q-PCR analyisis.
**Figure supplement 1.** Summary of our bulk RNA-Seq data.
**Figure supplement 2.** Post-sequencing quality check by using markers for multiple cell types.
**Figure supplement 3.** RNA in situ hybridization of SGN-specific genes.

down-regulated (FC >2) when transitioning to the next profiled age. For illustration, these transitions were represented in *Figure 3A* by the arrows between, for example, E15.5 and P1, or P1 and P8. In the case of 8047 genes, no significant differences were detected across the transitions and these genes were considered to show constant expression (----) (*Figure 3B*). By contrast, we identified 8 categories of dynamic genes, with different numbers (>10) of genes in each group (*Figure 3C–J*).

Next, to minimize any bias in our analysis, we used three different ontology annotations to examine enrichment of specific gene groups: GO (Gene Ontology) (*Ashburner et al., 2000*), PANTHER (*Mi et al., 2017*) and HUGO (*Braschi et al., 2019*) (*Figure 3K–M*). Among the 8047 genes that were constantly expressed in SGNs, we found that ribosomal proteins and spliceosome were enriched. The dynamic-expression group showed particular enrichment of genes encoding ion channels. For example, ion channel genes, such as *Hcn3*, *Kcnq3*, *Kcnt1* and *Cacna1d*, were transiently up-regulated between E15.5 and P1, and then rapidly down-regulated between P1 and P8, and these genes belonged to the gene category (ud–, 27 genes) as shown in *Figure 3H*. Accordingly, *Hcn3* is reported to be expressed in guinea pig SGNs and to exhibit an apical-to-basal expression gradient (*Bakondi et al., 2009*). It will be of interest to determine whether those transiently up-regulated genes are responsible for the spontaneous and sound-independent firing activity of SGNs that are only present before hearing onset (*Tritsch et al., 2007*).

*Mafb* belonged to the gene category (-d–) that was significantly down-regulated between P1 and P8 (*Figure 3E*), and this category included 226 genes. *Mafb* regulates the formation of large postsynaptic density containing abundant AMPA-type receptors/channels (*Yu et al., 2013*). In agreement with this finding, channel related genes and cell adhesion genes were enriched in those 226 genes, as suggested by all three ontology annotations (*Figure 3K–M*). Further investigation is required to determine whether other genes in this group regulate SGN functions.

In addition, deafness-associated genes were over-represented among the 371 genes that were up-regulated between E15.5 and P1 (u—), as illustrated in *Figure 3C and L*. Those deafness-associated genes included well-known HC genes, such as *Cdh23* (Cadherin-23) and *Cib2* (calcium and integrin binding family member 2), which were also detected in one previous single-cell SGN RNA-Seq study (*Shrestha et al., 2018*). Intriguingly, *Esrrb* (estrogen-related receptor beta) also belonged to this category and mutation of this gene causes autosomal-recessive nonsyndromic hearing impairment (autosomal recessive deafness-35, DFNB35) (*Collin et al., 2008*). Collectively, the results of our RNA-Seq analyses revealed roughly nine distinct expression patterns during SGN development. The categories with gene numbers less than 10 were not presented in *Figure 3*, but the entire patterns of each gene identified were listed in *Supplementary file 1*.

## *Scrt2* is specifically and constantly expressed in SGNs

To further validate the accuracy of our transcriptome data in vivo, we sought to exploit the specificity of the genes listed in *Figure 2* for generating knockin mouse strains. This approach allowed us to concurrently validate our findings, characterize genes at single-cell resolution, and develop valuable genetic tools for future SGN studies. Thus, we selected two candidate genes: *Scrt2*, from the category of SGN-specific genes showing a constant expression pattern and featuring a mean TPM value of 89.5 at P1 (*Figure 2—figure supplement 3D*); and *Celf4*, belonging to the category of SGN-specific genes showing a dynamic expression pattern and featuring a mean TPM value of 39.5 at P1 (*Figure 2—figure supplement 3H*).

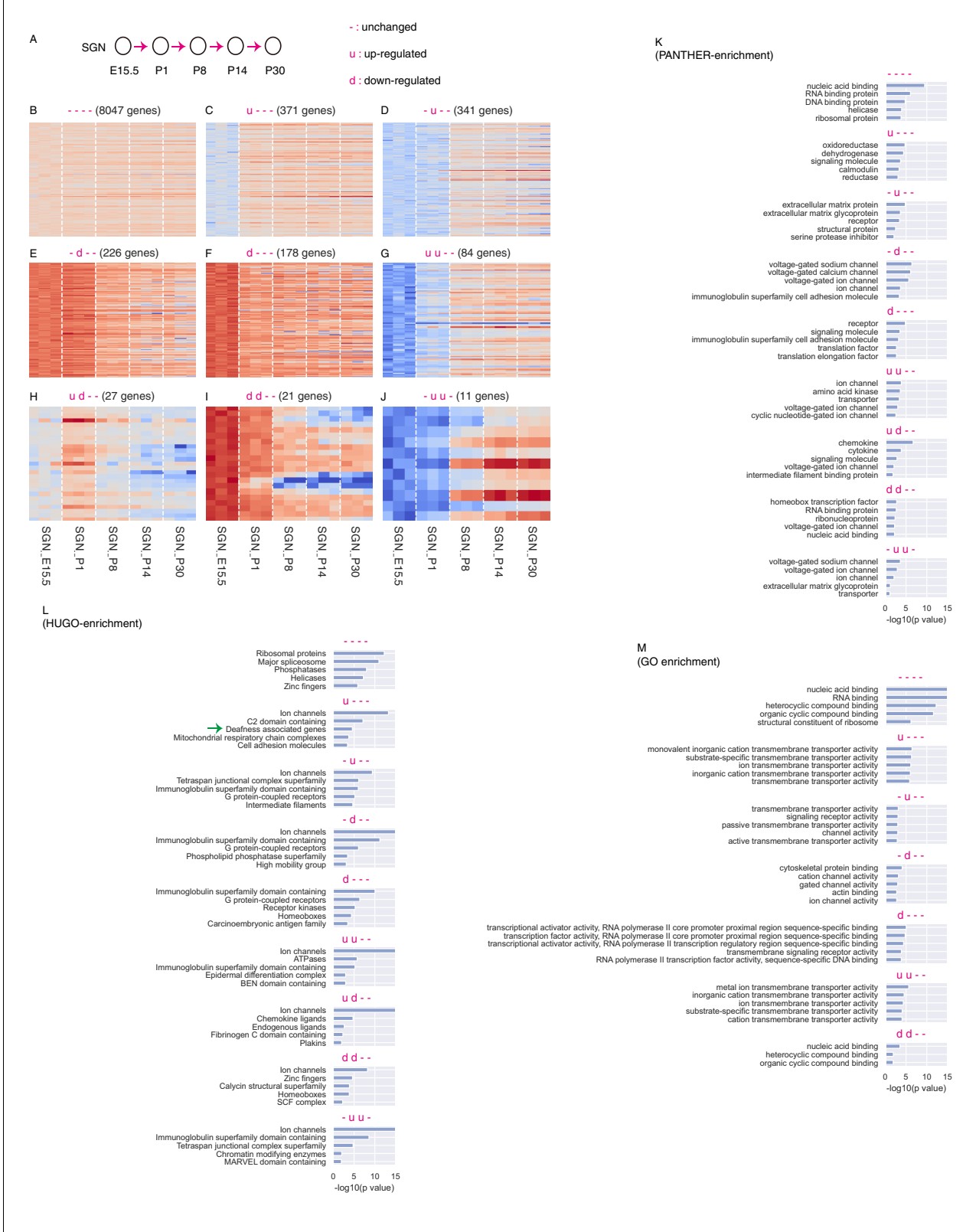

**Figure 3.** Computational analysis SGN genes without considering whether they are SGN-specific or not. (**A**) Cartoon depicting definition of up-regulation (u), down-regulation (d), and unchanged expression (-) during gene transitions between two neighboring ages (arrows). (**B**) The unchanged category included 8047 genes. (**C–J**) Dynamic genes were divided into eight categories. (**K–M**) Gene-group enrichment analysis was performed using three different gene annotations: HUGO, PANTHER, and Gene Ontology.

We selected *Scrt2* to generate a new knockin mouse strain: *Scrt2-P2A-tdTomato* (*Scrt2-tdTomato* for short). *Scrt2*, which encodes a protein named scratch-family zinc finger 2, has been found to control brain neurogenesis (*Paul et al., 2014*), but *Scrt2* expression in the inner ear has not been reported. We hypothesized that *Scrt2* might be involved in SGN development and thus selected this gene as a candidate. By using our usual CRISPR/Cas9 gene-targeting approach in zygotes (*Li et al., 2018*; *Zhang et al., 2018*), we inserted a P2A-tdTomato fragment immediately before the *TGA* stop codon of *Scrt2*, which should result in tdTomato expression being completely controlled by the endogenous *Scrt2* promoter and/or enhancer (*Figure 4A*). *Scrt2* expression was maintained intact by using the 2A-peptide approach (*Figure 4—figure supplement 1A–C*), as we have reported previously (*Li et al., 2018*). Southern blotting analysis confirmed that no random insertion of donor DNA in the mouse genome occurred (*Figure 4—figure supplement 1D and E*). Wild-type, heterozygous, and homozygous mice were identified through tail-DNA PCR-genotyping (*Figure 4—figure supplement 1F*). We used tdTomato as an indicator of *Scrt2* expression (at the mRNA level) in the analyses described below.

Although Sox2+/tdTomato+ cells were found in the hindbrain regions that were close to the otocyst (*Figure 4—figure supplement 2A–A''*), Sox2+ SGN progenitors, referred to as neuroblasts, did not express tdTomato at E10.5 (*Figure 4—figure supplement 2B*, n = 3, mouse numbers). This suggests that Scrt2 expression is not turned on in neuroblasts at E10.5. However, whole-mount analysis showed that many tdTomato+ cells were present in both SGN and VGN regions at E13.5 (white dotted lines in *Figure 4—figure supplement 2C*, n = 3, mouse numbers). Furthermore, triple labeling for tdTomato, Sox10 and Map2 demonstrated that tdTomato+ cells expressed the neuronal marker Map2 but not the glial marker Sox10 (*Figure 4—figure supplement 2D–E'''*). This confirmed that *Scrt2*, at least at the mRNA level, is expressed in SGNs at E13.5 (arrows in *Figure 4—figure supplement 2E–E'''*).

Similar patterns of tdTomato expression were observed in cochlear samples at E15.5 (n = 3, mouse numbers), by which time Myosin-VI+ HCs were present at basal and middle but not apical turns (*Figure 4B and C*). The Myosin-VI+ HCs did not express tdTomato (*Figure 4D*), demonstrating that *Scrt2* was not expressed in HCs. As expected, tdTomato+ cells were distributed densely in the SGN area in all three cochlear turns (*Figure 4E–E'''*), and the cells expressed Map2 but not Sox10 (arrows in *Figure 4F–F'''*). This again confirmed that *Scrt2* was expressed in SGNs but not glial cells.

We also analyzed inner ear samples of *Scrt2-tdTomato/+* mice at four postnatal ages: P1, P8, P14 and P30. Numerous tdTomato+ cells were present in SGN regions in all cochlear turns at P1 (n = 3, mouse numbers), as visualized in whole-mount (*Figure 4G*) and cryosection analysis (*Figure 4H*). As before, tdTomato+ cells expressed Map2 but not Sox10 (arrows in *Figure 4I–I'''*). Similar tdTomato expression patterns were also observed at P8, P14, and P30 (*Figure 4—figure supplement 3A–B''', C–D''' and E–F'''*, respectively; n = 3 mouse numbers/age). The tdTomato+ cells expressed Map2 but not Sox10 (arrows in *Figure 4—figure supplement 3B–B''' and D–D'''*) or Sox2 (another glial marker) (arrows in *Figure 4—figure supplement 3F–F'''*). With tdTomato serving as the read-out, the data overall indicate that *Scrt2* mRNA expression is turned on between E10.5 and E13.5 and then permanently and specifically maintained in SGNs. Glial cells and HCs did not express *Scrt2*.

## *Celf4* is dynamically expressed in both embryonic and adult SGNs

From the list of dynamically expressed genes (*Figure 2B*), we selected *Celf4* to confirm its temporal dynamics in vivo. RNA-Seq data analysis suggested that *Celf4* expression gradually increased from E15.5 to P30 (*Figure 2C*). This increment was further confirmed through qPCR analysis of *Celf4* (*Figure 2D*). *Celf4* (CUGBP Elav-like family member 4) encodes an RNA-binding protein involved in mRNA stability, protein translation, and seizure occurrence (*Ladd et al., 2001*; *Wagnon et al., 2012*; *Wagnon et al., 2011*). No previous study has reported *Celf4* expression in the inner ear. We selected *Celf4* because it was highly expressed at adult ages and was thus considered a promising candidate for generating a CreER driver line in which adult SGNs are specifically labeled within the inner ear. Moreover, because our previous work showed that RNA-binding proteins can precisely regulate gene translation and cell-fate maintenance (*Liu et al., 2015*), we also aim to characterize, in future studies, the potential roles of *Celf4* in inner ear SGN fate determination and differentiation.

By CRISPR/Cas9-mediated gene editing in zygotes, we generated the second knockin mouse strain, *Celf4-3xHA-P2A-iCreER-T2A-EGFP* (*Celf4-iCreER* for short), in which we inserted the fragment, *3xHA-P2A-iCreER-T2A-EGFP*, immediately before the *TGA* stop codon of the *Celf4* locus

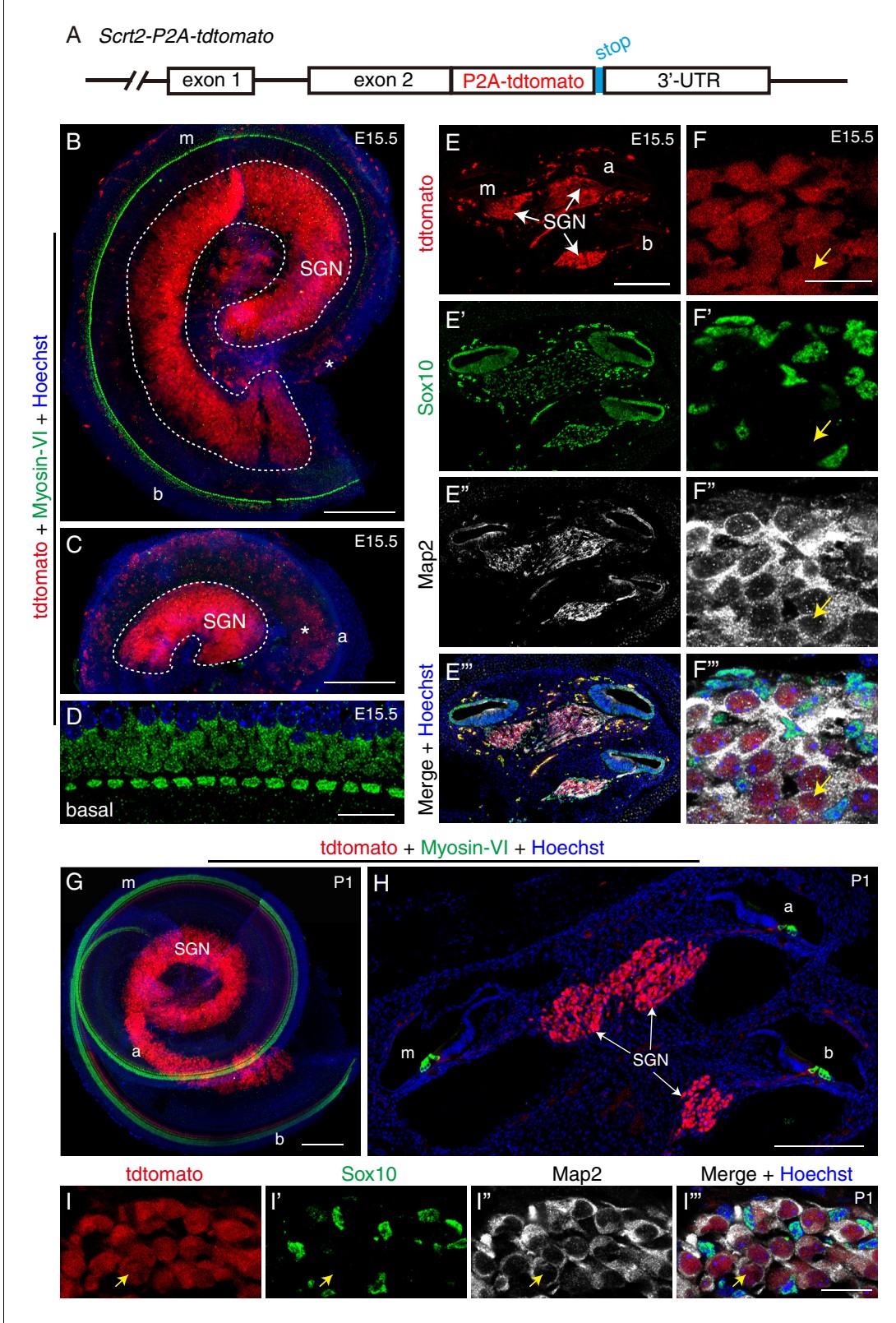

**Figure 4.** *Scrt2* is expressed in SGNs but not HCs or glia at E15.5 and P1. (**A**) Illustration of the genetically modified *Scrt2* locus. Detailed information please refer to *Figure 4—figure supplement 1*. (**B–C**) Double-labeling for HC-marker Myosin-VI and tdTomato in cochlear basal/middle portion (**B**) and middle/apical portion (**C**) dissected from *Scrt2-tdTomato/+* mice at E15.5. Several tdTomato+ cells were observed in the SGN area (white dotted circle). At this stage, the apical part did not yet harbor Myosin-VI+ HCs, and the middle turn contained only a single row of IHCs. Asterisk (**B and C**):

*Figure 4 continued on next page*

Figure 4 continued

blood cells showing high red autofluorescence. (D) High-resolution image of basal cochlea double-labeled for Myosin-VI and tdTomato. HCs did not express tdTomato. (E–F''') Low- and high-magnification (E–E''', F–F'') confocal images of cochlear tissues labeled for tdTomato, Sox10, and Map2 at E15.5. Arrows in (F–F'''): one tdTomato+ SGN that expressed Map2 but not Sox10. (G–H) Double-labeling for HC marker Myosin-VI and tdTomato in whole-mount (G) and cryosection (H) analyses at P1. Myosin-VI+ HCs did not express tdTomato (*Scrt2*). (I–I''') Triple-labeling for Map2, Sox10, and tdTomato. Arrows: one tdTomato+ SGN that expressed Map2 but not Sox10. This confirms that *Scrt2* was expressed in SGNs but not glial cells at P1. Scale bars: 200 $\mu$m (B, C, E, G, and H) and 20 $\mu$m (D, F, and I''').

The online version of this article includes the following figure supplement(s) for figure 4:

**Figure supplement 1.** Generation of *Scrt2-P2A-tdTomato* (*Scrt2-tdTomato*) knockin mouse strain.

**Figure supplement 2.** *Scrt2* is not expressed in neuroblasts at E10.5 but is expressed in SGNs at E13.5.

**Figure supplement 3.** *Scrt2* expression is maintained in SGNs at P8, P14, and P30.

(*Figure 5A* and *Figure 5—figure supplement 1A–C*). This mouse line enables three new assays: (1) precise visualization of Celf4 protein, because the protein C-terminus is tagged with 3 × HA (which can be detected using anti-HA antibodies); (2) lineage tracing of *Celf4*-expressing cells through tamoxifen injection at different ages by using iCreER; and (3) detection of *Celf4* mRNA expression based on EGFP. Absence of random insertion of donor DNA was confirmed through Southern blotting (*Figure 5—figure supplement 1D–E*), and knockin (*KI*) and wild type (+) alleles were identified using PCR genotyping (*Figure 5—figure supplement 1F*).

We first characterized inner ear samples from *Celf4-iCreER/+* mice at E15.5 (n = 3, mouse numbers). EGFP+ cells were observed in the SGN region of all three cochlear turns (*Figure 5B*). These EGFP+ cells expressed Map2 (arrows in *Figure 5C–C''*) but not Sox10 (arrows in *Figure 5D–D''*), confirming that these cells were SGNs and not glial cells. In agreement with our finding that *Celf4* mRNA levels gradually increased (dynamic expression) after E15.5 (*Figure 2C–D*), Celf4 protein was detected at a low signal-to-noise ratio at E15.5 with an anti-HA antibody (data not shown), but by P1, the signal-to-noise ratio of HA (Celf4) staining in EGFP+/Map2+ SGNs was considerably increased (arrows in *Figure 5F–F'''*). Moreover, EGFP and HA (Celf4) expression was maintained in Map2+ SGNs at P30 (*Figure 5G–H'''*). Furthermore, in addition to SGNs, VGNs expressed EGFP (*Figure 5B, E and G'''*). In summary, *Celf4* was expressed in SGNs and exhibited dynamic expression; this agrees with our RNA-Seq data (*Figure 2C*) and qPCR analysis of *Celf4* mRNA (*Figure 2D*).

## Fate mapping analysis by using *Celf4-iCreER/+* mouse strain

Besides analyzing the temporal pattern of *Celf4* expression, we performed fate mapping analysis by using the *Celf4-iCreER/+; Rosa26-CAG-loxp-stop-loxp-tdTomato* (Ai9)/+ mice (experimental group), as illustrated in *Figure 6A*. The control group included *Rosa26-CAG-loxp-stop-loxp-tdTomato/+* mice. Both groups (n = 3, mouse numbers in each) were injected with tamoxifen at P1/P2 and analyzed at P11 (*Figure 6B–D''*), or injected at P8/P9 and analyzed at P18 (*Figure 6E–G''*), or at P30/P31 and analyzed at P40 (*Figure 6H–J''*). No tdTomato+ cells were observed in any of the control groups. For brevity, those data were not presented here.

We detected tdTomato+ cells in SGN regions at P11, P18, and P40 at a similar density (white arrows in *Figure 6B,E and H*). These tdTomato+ cells expressed the SGN markers Tuj1 and Map2 (white arrows in *Figure 6C–C''*, *Figure 6F–F''* and *Figure 6I–I''*) but not the glial markers Sox10 and Sox2 (yellow arrows in *Figure 6D–D''*, *Figure 6G–G''* and *Figure 6J–J''*). Because tamoxifen was not taken up by all cells, a few Tuj1+ SGNs did not express tdTomato (asterisks in *Figure 6C–C''*). Quantification of the data did not reveal significant differences in the percentage of tdTomato+ SGNs between the different cochlear turns at P11 (*Figure 6K*), P18 (*Figure 6L*) or P40 (*Figure 6M*). At P11, P18 and P40, 33.5%–52.4%, 32.5–44.4% and 13.0–21.5% of SGNs were tdTomato+, respectively (*n* = 3, mouse numbers for each age). The EGFP signal here was also detected in cochlear duct cells, including HCs at P1 (*Figure 5E*); this should be regarded as background signal because no tdTomato+ HCs were visible in whole-mount analysis performed during the lineage tracing (tamoxifen injection at P1/P2) (*Figure 6—figure supplement 1*). We also detected tdTomato+ mesenchymal cells beneath the basilar membrane at P11 (*Figure 6—figure supplement 1D*).

To determine whether *Celf4* is expressed in the early embryonic otocyst, *Celf4-iCreER/+; Rosa26-CAG-loxp-stop-loxp-tdTomato/+* embryos (*n* = 3, embryo numbers) were administered tamoxifen at E10.5 and analyzed at E18.5 (*Figure 6—figure supplement 2*). Numerous tdTomato+ cells were

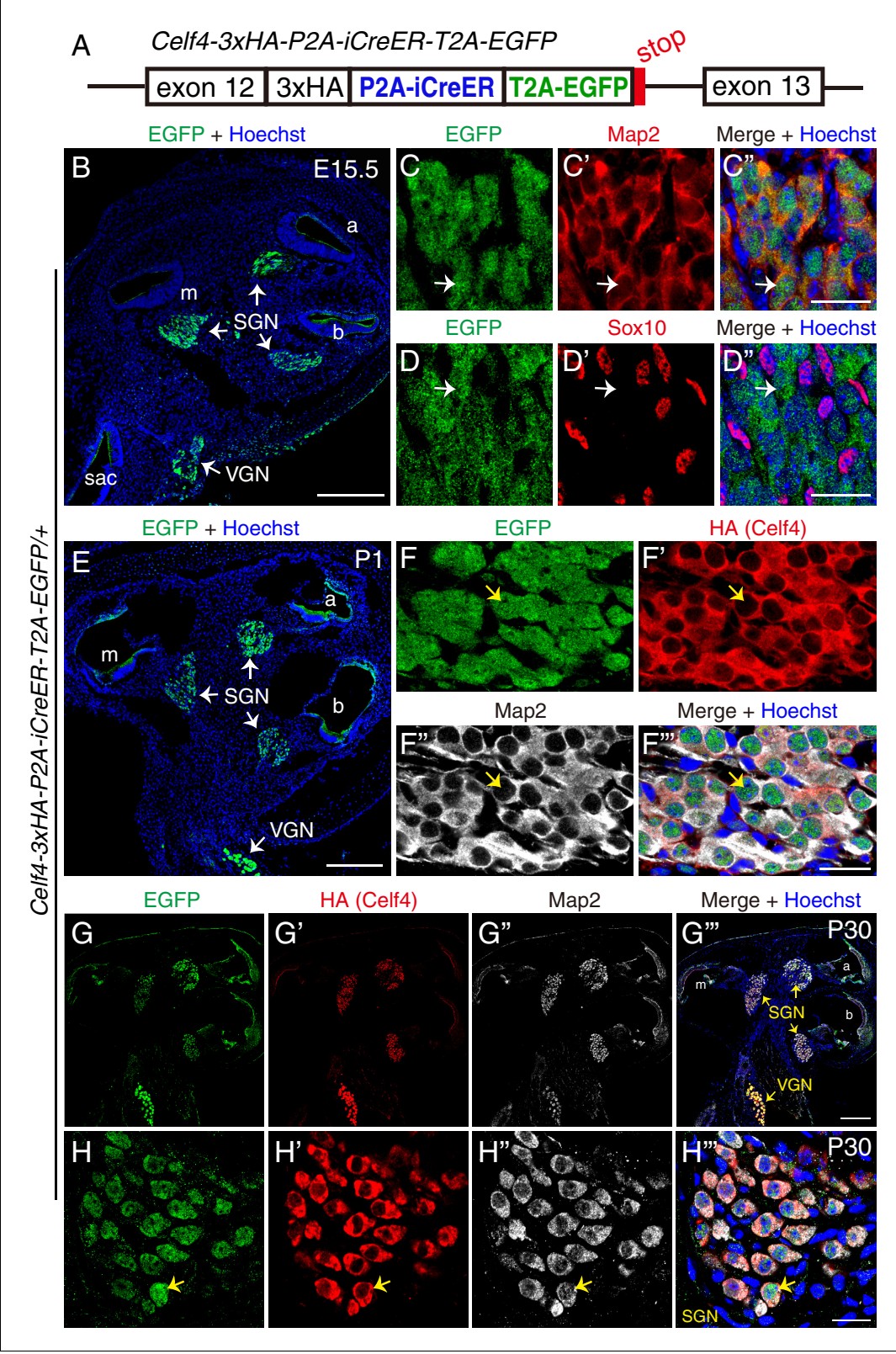

**Figure 5.** *Celf4* is detected in SGNs at E15.5 and further increased at P1 and P30. (**A**) Illustration of the genetically modified *Celf4* locus. Detailed information please refer to *Figure 5—figure supplement 1*. (**B**) Cochlear cryosectioned tissues from *Celf4-3xHA-P2A-iCreER-T2A-EGFP/+* (*Celf4-iCreER/+*) mice at E15.5 were stained with EGFP antibody. EGFP signal was observed in SGN areas in all three cochlear turns and in vestibular neuron (VGN)

*Figure 5 continued on next page*

*Figure 5 continued*

regions. (C–C'') Double-labeling for EGFP and neuronal marker Map2. Arrows: one EGFP+ cell that expressed Map2, confirming that it was an SGN. (D–D'') Double-labeling for EGFP and glial marker Sox10. Arrows: one EGFP + cell that did not express Sox10, confirming that it was not a glial cell. (E) Cochlear cryosectioned tissues from *Celf4-iCreER/+* mice at P1 were stained with EGFP antibody. (F–F''') Triple-labeling with EGFP, HA, and Map2 antibodies. Arrows: one SGN that expressed EGFP, HA-tagged Celf4, and Map2. (G–H'') Triple-labeling for EGFP, HA, and Map2 in cochlear cryosection tissues, shown at low (G–G''') and high resolution (H–H'''), from *Celf4-3xHA-P2A-iCreER-T2A-EGFP/+* (*Celf4-iCreER/+*) mice at P30. Arrows in (H–H'''): one SGN that expressed EGFP, HA (Celf4), and Map2. VGNs (G''') also expressed *Celf4*. Because the EGFP and HA signals were stronger in VGNs than in SGNs, *Celf4* expression level is expected to be higher in VGNs. Sac: sacculus. Scale bars: 200 $\mu$m (B, E and G''') and 20 $\mu$m (C'', D'', F''' and H''').

The online version of this article includes the following figure supplement(s) for figure 5:

**Figure supplement 1.** Generation of *Celf4-3xHA-P2A-iCreER-T2A-EGFP* (*Celf4-iCreER*) knockin mouse strain.

observed in SGN regions (*Figure 6—figure supplement 2A–C*). Those tdtomato+ cells expressed Mafb (arrows in *Figure 6—figure supplement 2D and D'*), but not Sox2 (arrows in *Figure 6—figure supplement 2E–E''*). Again, it confirms that tdTomato+ cells were SGNs but not glial cells. Interestingly, there was a significant difference regarding the percentage of tdTomato+ SGNs at basal (14.3 ± 1.53%), middle (5.04 ± 1.22%) and apical (0.60 ± 0.41%) turns (*Figure 6—figure supplement 2F*). It suggests that cells with higher *Celf4* expression around E10.5 preferentially develop into SGNs at basal or middle turns. Moreover, we analyzed cochlear samples of *Celf4-iCreER/+; Rosa26-CAG-loxp-stop-loxp-tdTomato/+* mice at P1 (n = 3, mouse number) that had not been administered tamoxifen. No tdTomato+ cells were observed in this case, which confirmed that Cre activity strictly depends on tamoxifen injection. Together, our data suggest that *Celf4* expression is turned on in neuroblasts (SGN progenitors) by E10.5 and maintained in SGNs at a high level in adult ages. *Celf4* was also expressed in other inner ear cells, such as mesenchymal cells (*Figure 6—figure supplement 1*).

## *Scrt2* and *Celf4* are expressed in both type I and type II SGNs at adult ages

In the adult cochlea, Type I SGNs account for ~95% of all SGNs and Type II for only ~5%. *Scrt2* and *Celf4* were expressed in numerous SGNs at P30 (*Figure 4—figure supplement 3E* and *Figure 5G–H'''*). Gata3 is highly expressed in adult Type II SGNs, but is not expressed or expressed at a very low level in adult Type I SGNs (*Nishimura et al., 2017*; *Yu et al., 2013*). To precisely determine whether *Celf4 and Scrt2* were expressed in Type II SGNs, we co-labeled for tdTomato and Gata3 in *Celf4-iCreER/+; Rosa26-CAG-loxp-stop-loxp-tdTomato/+* mice that were administered tamoxifen at P30/P31 and analyzed at P40, as well as in *Scrt2-P2A-tdTomato/+* mice at P30. In SGN regions of both models, tdTomato+/Gata3+ cells were observed (arrows in *Figure 7A–B''*), which showed that Type II SGNs expressed *Scrt2* and *Celf4*. The tdTomato+ cells exhibiting no or very low Gata3 expression were Type I SGNs (asterisks in *Figure 7A–B''*).

Besides SGN area (primarily in SGN peripheral region), Gata3 is widely expressed in many other cochlear cells including HCs (*Figure 7C*). Tdtomato+ fibers were found in OHC and IHC regions of *Celf4-iCreER/+; Rosa26-CAG-loxp-stop-loxp-tdTomato/+* mice at P40 (*Figure 7D–D'*). Again, it supported that *Celf4* was expressed both type I and II SGNs. However, it is also possible that some of those tdTomato+ fibers were efferent. Whole mount staining of tdTomato and Gata3 were also performed in cochlear samples of *Scrt2-P2A-tdTomato/+* mice at P30. Similar to *Figure 4—figure supplement 3E*, plenty of tdtomato+ SGNs were observed. Unfortunately, signal-to-noise ratio of tdtomato+ fibers close to IHCs and OHCs were low and the poor quality images were not presented here. We speculated that the expression level of tdtomato driven by *Scrt2* endogenous gene was lower than that driven by CAG promoter in *Rosa26* locus in *Celf4-iCreER/+; Rosa26-CAG-loxp-stop-loxp-tdTomato/+* mice. Together, we conclude that both Type I and Type II SGNs express *Scrt2* and *Celf4* at P30.

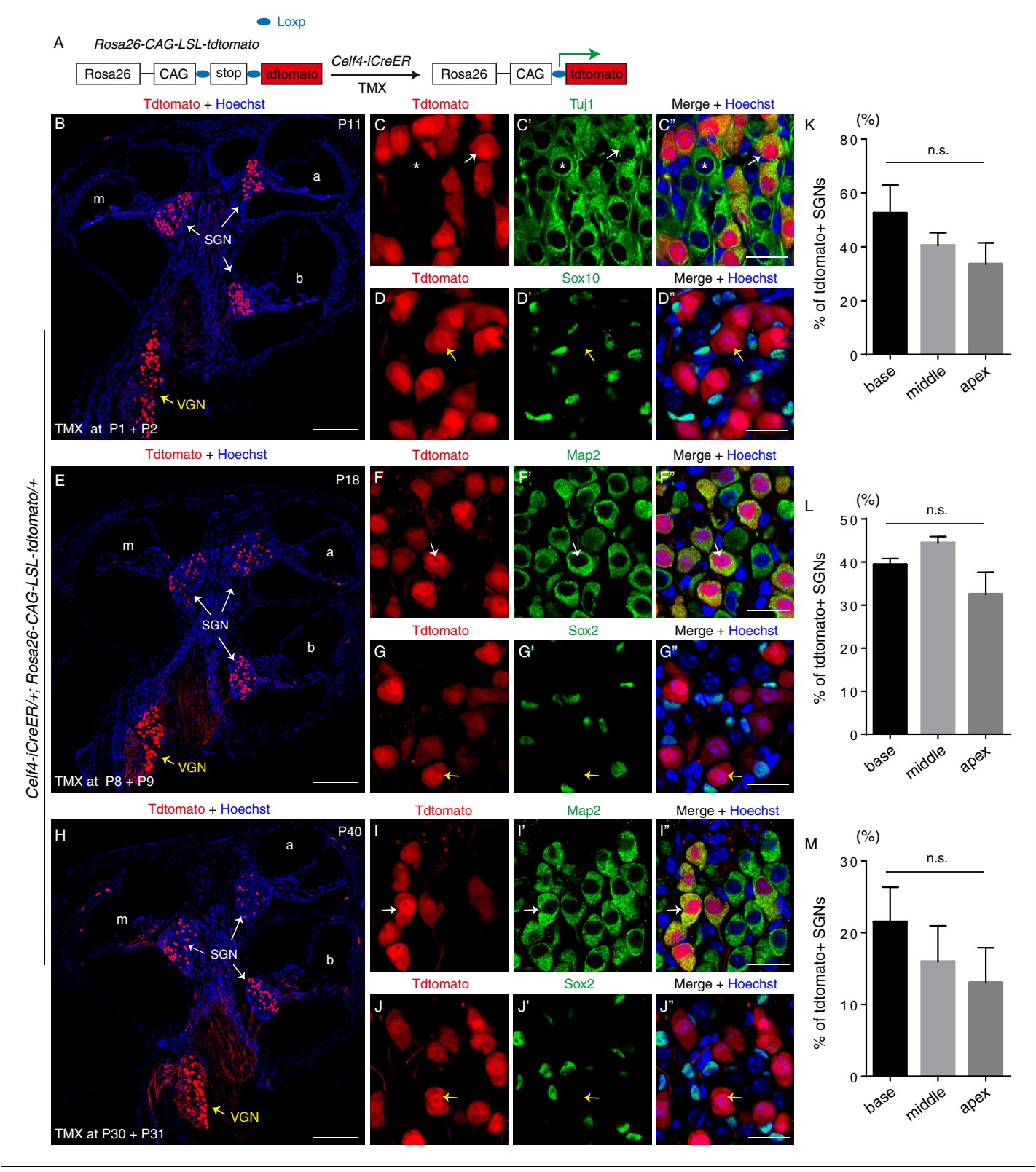

**Figure 6.** Fate-mapping analysis of *Celf4-iCreER/+; Rosa26-CAG-loxp-stop-loxp-tdTomato/+* mice across multiple ages. (**A**) Illustration of fate-mapping analysis method. (**B–D''**) Tamoxifen was injected at P1 and P2, and analysis was performed at P11. We detected tdTomato+ cells in SGN and VGN areas based on tdTomato endogenous fluorescence (**B**). Double-labeling was performed for tdTomato and Tuj1 (**C–C''**) or Sox10 (**D–D''**). White arrows in (**C–C''**): one Tuj1+/tdTomato+ SGN. Asterisk: one SGN that was not tdTomato+ due to the mosaicism of tamoxifen-mediated Cre-recombination.

*Figure 6 continued on next page*

Figure 6 continued

Yellow arrows in (D–D''): one tdTomato+ SGN that did not express Sox10. (E–G'') Tamoxifen was injected at P8 and P9, and analysis was performed at P18. As before, tdTomato+ cells were observed in SGN and VGN areas (E). Double-labeling was performed for tdTomato and Map2 (F–F'') or Sox2 (G–G''). Arrows in (F–F''): one Map2+/tdTomato+ SGN. Yellow arrows in (G–G''): one tdTomato+ SGN that did not express Sox2. (H–J'') Tamoxifen was injected at P30 and P31, and analysis was performed at P40. Again, most of the tdTomato+ cells were detected in SGN and VGN areas (H). Double-labeling was performed for tdTomato and Map2 (I–I'') or Sox2 (J–J''). Arrows in (I–I''): one Map2+/tdTomato+ SGN. As expected, the SGN indicated by yellow arrows in (J–J'') expressed tdTomato but not Sox2, confirming that glial cells did not express *Celf4*. (K–M) Quantification of tdTomato+ SGNs in each turn after tamoxifen injection at P1 and P2 (K), P8 and P9 (L), or P30 and P31 (M); no significant differences (n.s.) were found between different turns at each age. TMX: tamoxifen. Scale bars: 200 $\mu m$ (B, E, and H) and 20 $\mu m$ (C'', D'', F'', G'', I'', and J'').

The online version of this article includes the following source data and figure supplement(s) for figure 6:

**Source data 1.** Percentage of tdtomato+ SGNs when tamoxifen was given at postnatal ages.
**Figure supplement 1.** Besides SGNs, mesenchymal cells in the cochlea express *Celf4*.
**Figure supplement 2.** Fate-mapping analysis of *Celf4-iCreER/+; Rosa26-CAG-loxp-stop-loxp-tdTomato/+* mice at E10.5.
**Figure supplement 2—source data 1.** Percentage of tdtomato+ SGNs when tamoxifen was given at E10.5.

### *Scrt2* and *Celf4* are expressed in adult brain tissues

Last, we also briefly characterized the expression patterns of *Scrt2* and *Celf4* in the adult mouse brain. *Celf4* was previously reported to be expressed in the olfactory bulb, cortex, hippocampus and cerebellum in the adult brain (*Wagnon et al., 2011*; *Yang et al., 2007*; *Yue et al., 2014*). Accordingly, tdTomato+ cells were detected in these four brain areas of *Celf4-iCreER/+; Rosa26-CAG-loxp-stop-loxp-tdTomato/+* mice that were administered tamoxifen at P42 and P43, and analyzed at P52 (*Figure 7—figure supplement 1A–D*). Triple labeling for tdTomato, EGFP and HA revealed that both *Celf4* mRNA and protein were expressed (arrows in *Figure 7—figure supplement 1E–E'''*).

Similarly, tdTomato+ cells were also observed in the olfactory bulb, cortex, hippocampus and cerebellum of *Scrt2-P2A-tdtomato/+* mice at 4-month-old of age (*Figure 7—figure supplement 1F–I*). Notably, several tdTomato+ cells with long branched dendrites were observed in the cerebellum and these are likely Purkinje cells (arrow in inset of *Figure 7—figure supplement 1I*). The observed *Scrt2* expression in the adult cerebellum and cortex is consistent with the mouse Encode transcriptome data (*Yue et al., 2014*). *Scrt2* has been recently reported to be a critical regulator of cerebellum development (*Ha et al., 2019*). Future work is also needed to investigate whether *Celf4* and *Scrt2* are expressed in auditory ascending and descending neurons in brain. It will be able to address whether some tdTomato+ fibers were efferent or exclusively afferent (*Figure 7D–D'*). Together, our data suggest that *Celf4* and *Scrt2* are not exclusively expressed in SGNs and might be general regulators for neuron development.

## Discussion

We performed bulk RNA-Seq analysis of cochlear SGNs at five ages, E15.5, P1, P8, P14, and P30, covering distinct developmental states of SGNs. By comparing the transcriptomes of these SGNs and HCs/glial cells, we identified previously unknown SGN-specific genes, and we classified these genes as either constant or dynamic genes depending on their expression profile during development (*Figure 2A and B*). The accuracy of our dataset was further partially validated by our RNA in situ hybridization and thorough analysis of two knockin mouse strains that we generated. Thus, we have presented a precise and comprehensive dataset covering the transcriptomes of coding and noncoding genes that were not provided by previous studies.

### Comparison between our bulk RNA-Seq and three previous single-cell RNA-Seq analyses of SGNs

Pioneering single-cell RNA-Seq studies of SGNs were recently reported by three research groups (*Petitpré et al., 2018*; *Shrestha et al., 2018*; *Sun et al., 2018*). These groups used distinct methods: one group produced SGN cell suspensions from 9-week-old mice and used droplet microfluidics (10 × Genomics) (*Sun et al., 2018*); another group used manual picking and the Smart-seq2 method with P25–P27 mice (*Shrestha et al., 2018*); and the third group used fluorescence-activated cell sorting (FACS) to obtain SGNs from P17, P21, and P33 mice and constructed sequencing libraries by using Smart-seq2 (*Petitpré et al., 2018*). Despite differences related to age and single-cell RNA-Seq

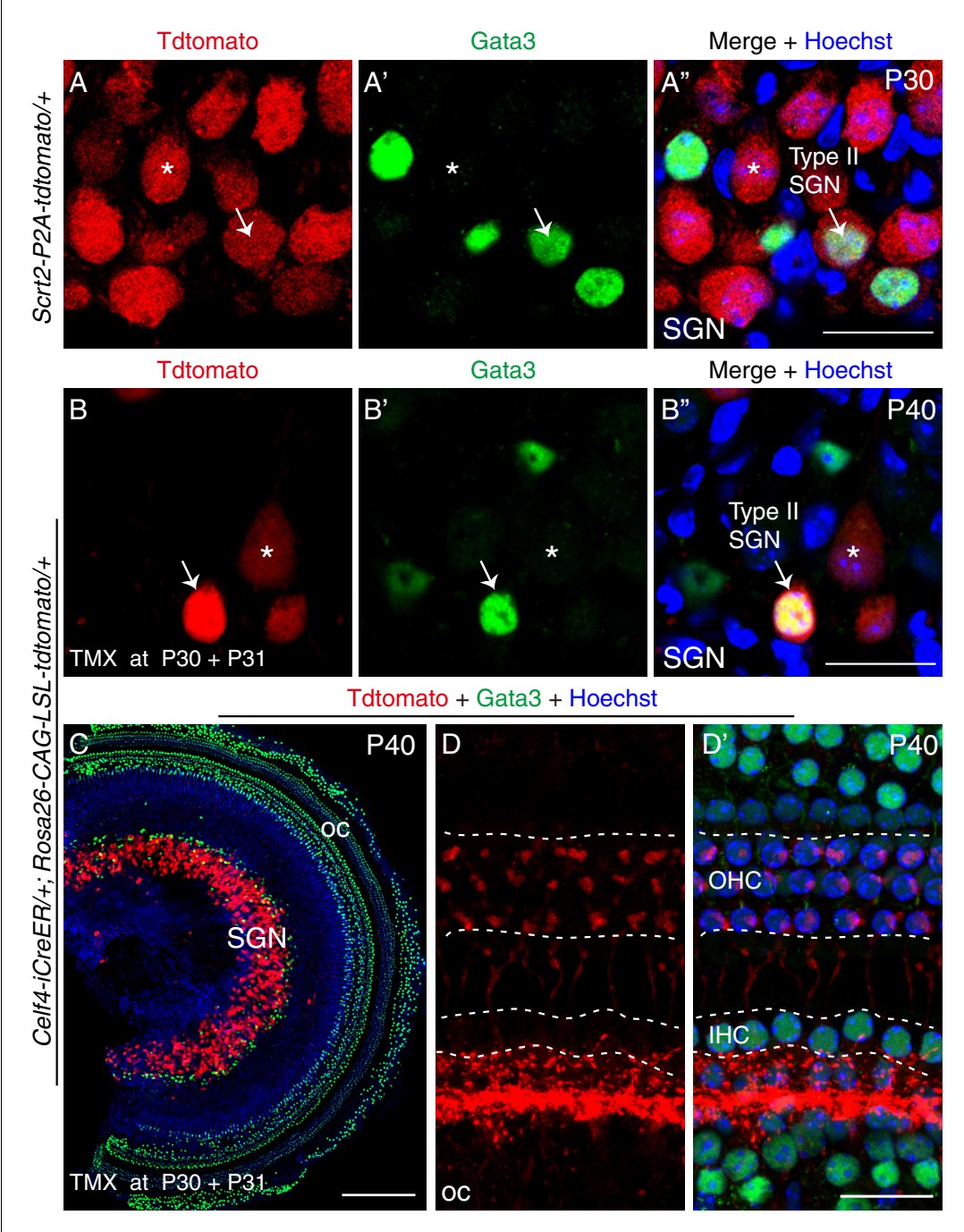

**Figure 7.** *Scrt2* and *Celf4* are expressed in both Type I and Type II SGNs in adult mouse cochlea. (**A–B''**) Co-labeling of tdTomato and Gata3 in cryosection cochlear samples from *Scrt2-P2A-tdTomato/+* mice at P30 (**A–A''**) and *Celf4-iCreER/+; Rosa26-CAG-loxp-stop-loxp-tdTomato/+* mice that were administered tamoxifen at P30 and P31 and analyzed at P40 (**B–B''**). Arrows: Type II SGNs, expressing both Type II marker Gata3 and tdTomato; asterisk: Type I SGNs, expressing only tdTomato; tdTomato represents expression of *Scrt2* (**A–A''**) and *Celf4* (**B–B''**). (**C**) Whole mount cochlear samples from *Celf4-iCreER/+; Rosa26-CAG-loxp-stop-loxp-tdTomato/+* mice were co-stained with tdTomato and Gata3. Gata3 was widely expressed in cochlea including cells in organ of Corti (oc) and SGN areas. (**D–D'**) High-resolution confocal z-stack projection images of oc region at P40. Tdtomato+ fiber terminals were distributed in both IHCs and OHCs. Scale bars: 200 $\mu$m (**C**), 20 $\mu$m (**A''**, **B''** and **D'**).

The online version of this article includes the following figure supplement(s) for figure 7:

**Figure supplement 1.** Scrt2 and Celf4 are also expressed in adult brain.

methods in the studies, similar conclusions were drawn: SGNs can be divided into Type Ia, Type Ib, Type Ic, and Type II subtypes. Multiple SGN genes, such as *Pou4f1*, *Lypd1*, *Calb1* and *Calb2*, are differentially expressed in each Type I subtype. Recently, nano-pore long-read RNA-Seq analysis has revealed the splicing diversity of inner ear HCs and Deiters'cells (*Ranum et al., 2019*); we speculate that a higher degree of heterogeneity (than currently appreciated) will be found among SGNs if long-read RNA-Seq is used.

All three aforementioned single-cell RNA-Seq studies focused on the heterogeneities among SGNs or among different cochlear turns for tonotopic analysis. By contrast, we focused here on comparing the transcriptomes of SGNs and two other inner ear cell types, HCs and glia. Our aim was to identify genes that were expressed specifically in SGNs within cochlea and exhibited either constant (e.g. *Scrt2*) or dynamic (e.g. *Celf4*) expression patterns. We expect these genes to represent promising candidate regulators of SGN cell-fate determination and/or differentiation. Accordingly, *Mafb* appeared in our list of dynamic genes, and this gene is known to be required for SGN to function properly (*Yu et al., 2013*). Because only HCs and glial cells were used as reference cells in this study, we cannot rule out the possibility that other cell types in the cochlea also transiently or permanently express the genes listed in *Figure 2*. In accord with this view, *Celf4* expression was also sporadically detected in mesenchymal cells in our lineage-tracing assays (*Figure 6—figure supplement 1*). Comprehensive analysis in the future will be necessary to confirm the in situ expression of each of these genes.

Another distinguishing feature of our study as compared with other studies is that we did not limit our analysis to coding mRNAs. Whereas single-cell RNA-Seq can only capture mRNAs harboring poly-A tails, our approach uses both oligo-dT and random primers and thus captures both mRNAs and noncoding RNAs. For example, *mir680-2* (microRNA 680–2) was identified as an SGN dynamic gene (*Figure 2B*). Our method also allowed for greater depth of coverage than before because we pooled ~100 pure SGNs; this led to the detection of ~15,000 genes (TPM >1), which is considerably deeper than the coverage in previous SGN single-cell RNA-Seq studies:~2000 genes with 10 × Genomics v2 (*Sun et al., 2018*) or ~8000 genes with Smart-seq2 approach (*Shrestha et al., 2018*). It is reported that SGNs were easily contaminated by blood cells (*Sun et al., 2018*). We found that genes of blood cells, such as *Hbb-bs* and *Hbb-bt*, were depleted in our samples (*Figure 2—figure supplement 2*), which indicated that the three SGN washing steps in our procedure markedly diminished the RNAs released by blood cells. In summary, to our knowledge, this bulk RNA-Seq database represents the first report of SGN-specific genes showing constant or dynamic expression and it features a large and vast covering depth. A combination of our bulk RNA-Seq and previous single-cell RNA-Seq of SGNs will enable future investigation of SGN gene expression from multiple perspectives. Last, besides the aforementioned RNA-Seq data of SGNs, data are available from one microarray study of SGNs and VGNs covering ages from E12 to P15 (*Lu et al., 2011*). Interestingly, both the microarray data and our bulk RNA-Seq data show that immune-related genes are overrepresented in SGNs, and thus the functions of these genes warrant further investigation. Different from microarray data, our bulk RNA-Seq can reveal SGN-specific genes because we used HCs and glias as reference. In addition, relative to microarray analysis, RNA-Seq dataset can reveal wider range of gene expression level difference.

## Applications of *Scrt2-tdTomato/+* knockin mouse model

We generated the knockin *Scrt2-tdTomato/+* mouse strain not only because we sought to validate our RNA-Seq data, but also to provide a useful genetic tool for future studies on SGN development and regeneration. For example, attempts to regenerate SGNs from neighboring glial cells are of increasing interest and importance (*Noda et al., 2018*); but conducting functional patch-clamp studies to validate such attempts requires the precise identification of new SGNs without antibody staining. Therefore, use of a fluorescent protein whose expression is driven by an SGN-specific gene promoter/enhancer can facilitate such studies. For this purpose, *Scrt2* is more suitable than *Parvalbumin* and *Calbindin*, which are expressed both in HCs and SGNs. Previous single-cell RNA-Seq data (*Shrestha et al., 2018*) and our own *Scrt2* analysis data showed that *Scrt2* is expressed in both Type I and Type II SGNs (*Figure 7A–A''*). Thus, regardless of whether new SGNs resemble the Type I or II SGN subtype, the *Scrt2-tdTomato/+* strain can be used as a readout.

Our *Scrt2-tdTomato/+* strain also offers advantages over *Mapt (Tau)-EGFP/+* mice (Jax#: 029219). First, *Mapt-EGFP/+* are knockout mice and EGFP expression replaces Mapt expression

(*Tucker et al., 2001*). However, because we used the 2A-peptide approach to generate *Scrt2-tdTomato/+* mice, Scrt2 expression is intact even in the homozygous background for brighter tdTomato. Second, the EGFP pattern in *Mapt-EGFP/+* mice primarily recapitulates the *Tuj1* (β-tubulin III) pattern. Tuj1 is reported to be only strongly expressed in Type I SGNs at adult ages (*Nishimura et al., 2017*). Therefore, if the goal is to label as many SGNs as possible with a bright fluorescent protein, particularly from the SGN regeneration perspective, the *Scrt2-tdTomato/+* strain is comparatively more powerful. Note that tdTomato signal was strong in cell body of SGNs but weak in their fiber terminals.

## Potential contributions of *Celf4-iCreER/+* knockin mouse strain to future SGN studies

The Cre/loxP system has been widely used in genetic studies in the inner ear field in the past two decades (*Cox et al., 2012*). Multiple Cre mouse strains are available, especially in the case of HCs. *Gfi1Cre* (knockout) and *Gfi1-P2A-GFP-CreERT2 (Gfi1-GCE)* knockin mice can be used to perform conditional loss-and-gain of function studies in HCs soon after HCs are formed (*Tang et al., 2019*; *Yang et al., 2010*). In the transgenic strain *Atoh1-CreER+*, neonatal cochlear HCs are targeted (*Chow et al., 2006*). By contrast, knockin strains *Prestin-CreER/+* and *vGlut3-iCreER/+* permit genetic manipulations in cochlear OHCs and IHCs at different postnatal ages (*Fang et al., 2012*; *Li et al., 2018*). These powerful tools together allow temporal genetic manipulations to be performed on HCs or specific HC subtypes.

Comparatively fewer SGN-specific CreER tools are currently available. Besides *ShhCre/+* (used in this study), *Neurog1-Cre/+* and *Bhlhe22-Cre/+* lines (*Quiñones et al., 2010*; *Ross et al., 2010*), are highly suitable tools for deleting or activating genes in embryonic SGNs (*Appler et al., 2013*; *Yu et al., 2013*). However, these are all straight Cre lines, and gene deletion or activation occurs soon after SGN formation without temporal control with tamoxifen. A *Neurog1-CreERT2* transgenic line (Jax#: 008529) is also available, and this can be used for embryonic tamoxifen-mediated temporal control (*Koundakjian et al., 2007*; *Raft et al., 2007*). However, in our RNA-Seq analysis, *Neurog1* expression was rarely detected in SGNs at E15.5 and the tested postnatal ages, which suggests that *Neurog1* is turned off after late embryonic ages. Therefore, none of the aforementioned lines can be used to manipulate gene functions starting at postnatal ages and particularly at adult ages.

Unlike the Cre lines mentioned above, *Celf4-iCreER/+* is not very sensitive to the time of tamoxifen injection, and can efficiently target SGNs when tamoxifen is injected at P1/P2, P8/P9, or P30/P31 (*Figure 6*). We also observed tdTomato+ SGNs when tamoxifen was administered in *Celf4-iCreER/+; Rosa26-CAG-loxp-stop-loxp-tdTomato/+* mice during embryonic stages (E10.5). Thus, *Celf4-iCreER/+* holds the potential to emerge as a highly valuable tool for temporally manipulating SGN gene function at different ages. Besides *Celf4-iCreER/+*, a BAC transgenic *Mafb-CreER+* line is available that can be used for sparse labeling of SGNs with tdTomato at P25–P27, assuming that tamoxifen is injected around P20 or earlier (*Di Meglio et al., 2013*; *Shrestha et al., 2018*). Mafb expression in Type I SGNs gradually decreased during postnatal ages (*Yu et al., 2013*). *Mafb* expression in Type I SGNs was substantially lower than in Type II SGNs. Therefore, we expected the percentage of Type I SGNs labeled in the *Mafb-CreER+* strain to be smaller than that of Type II SGNs at adult ages. However, our data showed that Type II SGNs were frequently labeled by *Celf4-iCreER/+* at P30 (*Figure 7B–B''*). All of the Cre or CreER lines described above, except *Prestin-CreER/+*, are also active in the brain and other tissues. We are not aware of any SGN-specific Cre or CreER line that is exclusive for SGNs and shows no expression in any other mouse tissues.

## What are the functions of SGN-specific genes showing constant or dynamic expression?

We noticed that many of the genes identified in our study were not previously known to be SGN-specific within cochlea. Both *Scrt2* and *Celf4* were expressed in both type I and II SGNs (*Shrestha et al., 2018*). *Mafb*, one of the genes found here to show dynamic expression, is recognized to control SGN differentiation and normal synaptogenesis between SGNs and IHCs (*Yu et al., 2013*). Apart from this example, the function of the majority of genes identified in SGNs remains unknown. One of our SGN-specific genes, *Esrrg*, which encodes estrogen-related receptor-γ, is reported to be involved in congenital hearing loss, but the detailed mechanisms underlying its

function remain unclear (*Schilit et al., 2016*). Another SGN-specific gene, *Lrrc55* (leucine-rich-repeat containing 55), is detected in SGNs as early as E15.5. Intriguingly, *Lrrc52* (leucine-rich-repeat containing 52) is selectively expressed in Type I SGNs (*Shrestha et al., 2018*). *Lrrc52*, *Lrrc55*, and other members of this gene family (*Lrrc26* and *Lrrc58*) are involved in the regulation of BK channels (large-conductance calcium- and voltage-activated potassium channels) and in various acoustic information-processing steps (*Pyott and Duncan, 2016*). For detailed functional validation of these newly identified SGN-specific genes, in vivo genetic studies are necessary. This will be one of our future research directions, and it will be facilitated by our recently established high-throughput CRISPR/Cas9-mediated base-editing (loss-of-function) method (*Zhang et al., 2018*).

In summary, we have provided a comprehensive high-quality SGN transcriptome database covering five ages, from early embryonic stage (E15.5) to the adult stage (P30) in mice, with deep coverage of a large number of genes. In contrast to other RNA-Seq analyses of SGNs, our study focused on identifying SGN-specific genes exhibiting constant or dynamic expression. Moreover, we generated two knockin mouse strains here: *Scrt2-tdTomato/+* and *Celf4-iCreER/+*; we believe these will serve as useful tools for future studies on inner ear SGNs and on the central nervous system in general.

# Materials and methods

**Key resources table**

| Reagent type (species) or resource | Designation | Source or reference | Identifiers | Additional information |
|---|---|---|---|---|
| Strain, strain background (*Mus musculus*) | *ShhCre/+* | *Liu et al., 2010* | Jackson Lab | stock #: 005622 |
| Strain, strain background (*Mus musculus*) | *Atoh1-CreER+* | *Chow et al., 2006* | MMRRC | stock #: 029581-UNC |
| Strain, strain background (*Mus musculus*) | *vGlut3-iCreER/+* | *Li et al., 2018* | Available upon request from Liu Lab | Please contact (zhiyongliu@ion.ac.cn) |
| Strain, strain background (*Mus musculus*) | *Scrt2-P2A-tdTomato/+* | It is a knockin mouse line where tdTomato can reflect *Scrt2* mRNA expression pattern | Jackson Lab | stock #: 034390 (will be deposited soon) |
| Strain, strain background (*Mus musculus*) | *Celf4-3xHA-P2A-iCreER-T2A-EGFP/+* | It is a knockin mouse line where HA is tagged at Celf4 protein c terminus. In addition, iCreER and EGFP are controlled by *Celf4* promoter/enhancer | Jackson Lab | stock #: 034391 (will be deposited soon) |
| RNA extraction kit | PicoPure RNA Isolation Kit | Thermo Scientific | Cat#: KIT0204 | Extracting RNA from isolated cells |
| cDNA generation kit | Ovation RNA-Seq V2 | Tecan Genomics | Cat#:7102–32 | Converting RNA to cDNA |
| Library construction kit | Ovation Rapid DR Multiplex System | Tecan Genomics | Cat#: 0319–32 | Generating sequencing library with eight samples multiplexing |
| Digestion enzyme | Protease | Sigma | Cat#: P5147 | Digesting inner ear tissues |

## Genetic labeling and pre- and post-sequencing quality checks of SGNs, HCs, and glial cells, and library construction

SGNs at different ages were manually picked to guarantee the highest purity of our samples. Manual picking allowed us to not only monitor the tdTomato fluorescence, morphology, and health of the cells, but also avoid picking up cellular debris. The picked cells were washed three times, to

maximize sample purity, before loading into lysis buffer. We also used manual picking for one reference-cell population, P8 glial cells. We initially attempted to use FACS to obtain glial cells, but the purity of these cells did not meet our quality criteria. In the case of the second reference-cell population, P12 HCs (both cochlear and vestibular), the samples obtained using FACS met our quality criteria and were thus used.

Three genetic models were used for labeling each cell type with tdTomato. SGNs were genetically labeled with tdTomato by using *ShhCre/+; Rosa26-CAG-loxp-stop-loxp-tdTomato/+* strain (*Figure 1*). After microdissection, cochlear tissues were examined under a fluorescence microscope to ascertain whether tdTomato+ cells were limited to the SGN region; this step eliminated any sample exhibiting germ-line leakage of *ShhCre/+*: leakage results in all cells in the inner ear becoming tdTomato+. HCs were genetically labeled with tdTomato by using *Atoh1-CreER+; Rosa26-CAG-loxp-stop-loxp-tdTomato/+* mice that were administered tamoxifen at P0 and P1 (*Chow et al., 2006*). Lastly, glial cells and cochlear IHCs were designed to express tdTomato+ by using *vGlut3-iCreER/+; Rosa26-CAG-loxp-stop-loxp-tdTomato/+* mice that were administered tamoxifen at P2 and P3 (*Li et al., 2018*), and only the tissues in the SGN area were dissected out in order to discard tdTomato+ IHCs. We followed the same tissue-digestion and cell-picking protocols as previously described (*Li et al., 2018*; *Liu et al., 2015*). PicoPure RNA Isolation Kit (KIT0204, Thermo Scientific) was used to extract RNA. Ovation RNA-Seq V2 (7102–32, Tecan Genomics) and Ovation Rapid DR Multiplex System (0319–32, Tecan Genomics) were used for cDNA conversion and library construction.

The pre-sequencing quality check was performed using qPCR. *Mafb*, *Sox10*, and *Myosin-VI* were selected as genes specific for SGNs, glia, and HCs, respectively, and the criterion applied was that for each cell type, the sample must show at least 30-fold enrichment or depletion in *Mafb*, *Sox10*, and *Myosin-VI*, as exemplified in P8 SGNs (*Figure 1D-1F*). The qPCR data were statistically analyzed using Student's *t* test. *Supplementary file 2* includes the qPCR primer sequences for each gene. The post-sequencing quality check was performed using additional known genes to verify the quality and purity of the cells. Besides SGN, HC, and glial markers, we evaluated genes covering seven other cell types: inhibitory neurons, catecholaminergic neurons, oligodendrocytes, astrocytes, microglia, choroid plexus cells, and blood cells—as well as housekeeping genes (*Figure 2—figure supplement 2*). Only samples passing both pre-and post-sequencing checkpoints were included in our study. All mice were bred and raised in SPF-level animal rooms, and animal procedures were performed according to the guidelines (NA-032–2019) of the IACUC of Institute of Neuroscience (ION), Chinese Academy of Sciences.

## Computational analysis of RNA-Seq data

RNA-Seq reads in FASTQ files were mapped to the mouse mm10 genome using STAR (*Dobin et al., 2013*). Mapped reads were then quantified with featureCounts (*Liao et al., 2014*) using the mouse gencode.vM15 annotation (*Harrow et al., 2012*). Inter-group pairwise differential expressions were calculated: genes featuring CPM (counts per million) values of >1 for all the replicates in at least one group were selected to be considered for the subsequent analysis (21,293 out of 52,550, including HC and glia, or 17, 531 excluding HCs and glia).

To filter out SGN-specific genes showing either constant or dynamic expression (*Figure 2A and B*), the count data were TMM (the weighted trimmed mean of M-values) normalized and pairwise differential expressions were evaluated for all the pairs of groups using the limma package with its voom method (*Law et al., 2014*). First, SGN-specific genes showing constant expression were defined as genes at the intersection of these four conditions (*Figure 2A*): 1) differentially expressed between all SGNs and HC, glia (with fold change, FC > 4 and qval <0.01); 2) not differentially expressed between SGNs at different ages (FC < 2), 3) TPM (transcripts per million) values in HC and glia populations were <5; 4) mean TPM value in SGNs was >30. Second, SGN-specific genes showing dynamic expression were extracted according to the following conditions (*Figure 2B*): 1) a significant difference was found between at least one of the SGN groups and both HC and glia (qval <0.01 and FC > 4); 2) a significant difference was present in at least one paired comparison between different SGN groups (qval <0.05 and FC > 2); 3) maximum mean TPM of SGN group was >75; 4) maximum mean TPM of SGN group was at least 25 times as much as that of HC and glia. The use of relatively less stringent criteria should result in the identification of additional SGN-specific genes showing constant or dynamic expression patterns.

We also analyzed dynamics of SGN genes (CPM > 10 for all replicates in at least one age; 9349 genes) without considering their expression patterns in HC and glia (*Figure 3*). To classify gene dynamics, they are designated as up-regulated (u), down-regulated (d) or unchanged (-), between one age and the next age (e.g. SGN_E15.5 to SGN_P1). Up or down-regulation is determined based on these values: qval <0.05 and FC >2. Unchanged is determined based on qval >0.1, FC <2 and maximum element-wise FC <4. All other altered genes were designated as 'noisy'. For simplicity, the noisy genes were included in the unchanged group (*Figure 3B*). The dynamics for the entire period were classified as the combination of the individual transitions (e.g. 'u—" indicates up-regulation from E15.5 to P1, following by unchanged expression from P1 to P8, P8 to P14, and P14 to P30). Note that the dynamics here were focused on changes between two neighboring ages (*Figure 3C–J*). This differs from the dynamics in the context of SGN-specific genes showing dynamic expression patterns, which were defined as genes that exhibited significantly different expression between any of the two compared ages (*Figure 2B*). Because of this difference, *Celf4* belonged to unchanged (——) category (*Figure 3B*). However, *Celf4* expression at the adult age (P30) was significantly higher than that at E15.5 (*Figure 2C*), as also validated by qPCR analysis of *Celf4* (*Figure 2D*). Therefore, *Celf4* was defined as an SGN-specific gene showing dynamic expression (*Figure 2B*). Dynamic patterns of each gene (*Figure 3B–J*) were described in *Supplementary file 1*.

Last, to find what kind of genes belong to each of the above classified group, gene enrichment analyses (*Figure 3K–M*) were performed for each dynamic gene group using 1) the top-level HUGO gene groups (*Braschi et al., 2019*), 2) PANTHER Classification System (*Mi et al., 2017*), and 3) the Molecular Function component of the Gene Ontology Annotation (*Ashburner et al., 2000*). Enrichment *p*-value was calculated using hypergeometric distribution. All the raw data of our RNA-seq have been deposited in GEO (Gene Expression Omnibus) under the accession: GSE132925. The read counts of all the genes are available in the file: GSE132925-cochlear-SGN-cnts.txt.gz attached in GEO.

## Generation of *Scrt2-tdTomato/+* and *Celf4-iCreER/+* knockin mouse strains

The *Scrt2-tdTomato/+* mouse was generated by co-injecting a pre-tested sgRNA against *Scrt2*, donor DNA (*Figure 4—figure supplement 1*), and Cas9 mRNA into one-cell-stage mouse zygotes. The *Scrt2* sgRNA sequence used was *5'-Gcctcggcgggcatcccgca-3'*. Detailed donor DNA sequences are available upon request. Junction PCR was used to screen F0 mice, after which the F0 mice were crossed with wild-type C57BL/6 mice for germ-line transition and to produce F1 mice. The F1 mice were further screened using junction PCR and Southern blotting (*Figure 4—figure supplement 1D and E*). The Southern blotting results confirmed that no random insertion of donor DNA occurred in the mouse genomes; Southern blotting was performed according to our previously described protocol (*Li et al., 2018*). Tail-DNA PCR allowed us to distinguish wild-type (+/+), heterozygous (*KI/+*), and homozygous (*KI/KI*) mice. When primers F2 and R1 were used, the obtained KI PCR amplicon was 894 bp long; conversely, with primers F1 and R1, the obtained wild-type PCR amplicon was of 350 bp long. Theoretically, primers F1 and R1 should generate a 1844 bp band in the KI allele, but because of the short extension time of our PCR protocol, this band was not produced.

To produce the *Celf4-iCreER/+* strain, a pre-tested *Celf4* sgRNA, donor DNA, and Cas9 mRNA were injected into one-cell-stage mouse zygotes. All other procedures were identical to those described above and thus are not repeated here. We selected the gene *Celf4-201* (ENSMUST00000025117.13 in Ensembl website) for gene targeting. *The Celf4 sgRNA sequence used was 5'-tacgggcgattggcgtcttt-3'.* Southern blotting analysis ruled out random insertion of donor DNA into the mouse genomes. When primers F1 and R1 were used, the obtained wild-type PCR amplicon was 471 bp long; with primers F2 and R1, the KI PCR amplicon was 612 bp long. As before, primers F1 and R1 did not successfully produce a 3402 bp amplicon in the KI allele due to the limited extension time of our PCR protocol. All sequences used for the genotyping-PCR primers for the two mouse strains are listed in *Supplementary file 2*. These two strains will be available at The Jackson Laboratory Repository (http://jaxmice.jax.org/query) under JAX Stock Nos. 034390 and 034391.

## Sample processing, histology and immunofluorescence and RNA in situ hybridization

For lineage-tracing analysis of *Celf4-iCreER/+; Rosa26-CAG-loxp-stop-loxp-tdTomato /+* mice, tamoxifen (T5648, Sigma) dissolved in corn oil (C8267, Sigma) was administered at P1/2 or P8/9 at a dosage of 3 mg/40 g body weight, or at P30/31 at a dosage of 9 mg/40 g body weight. Inner ear tissues from embryonic mice or mice younger than P10 were dissected out and fixed in fresh 4% PFA overnight at 4℃. Mice older than P10 were heart-perfused with 1 × PBS and then perfused with fresh 4% PFA, and this was followed by a second fixation in fresh 4% PFA overnight at 4℃. Inner ear tissues were washed three times with 1 × PBS and then either directly used in whole-mount analysis or soaked in 30% sucrose at 4℃ overnight, embedded in OCT, cryosectioned at 14 μm thickness, and then examined.

The following primary antibodies were used: anti-Myosin-VI (rabbit, 1:200, Cat#: 25–6791, RRID: AB_10013626, Proteus Bioscience), anti-Mafb (rabbit, 1:300, Cat#: HPA005653, RRID:AB_1079293, Sigma), anti-GFP (chicken, 1:1000, Cat#: ab13970, RRID:AB_300798, Abcam), anti-Sox2 (goat, 1:1000, Cat#: sc-17320, RRID:AB_2286684, Santa Cruz Biotechnology), anti-HA (rat, 1:200, Cat#: 11867423001, RRID:AB_390918, Sigma), anti-Tuj1 (mouse, 1:500, Cat#: 801201, RRID:AB_2313773, BioLegend), anti-Gata3 (goat, 1:200, Cat#:AF2605, RRID: AB_2108571, R and D systems), anti-Map2 (rabbit, 1:400, Cat#: M3696, RRID:AB_1840999, Sigma), and anti-Sox10 (goat, 1:200, Cat#: sc-17342, RRID:AB_2195374, Santa Cruz Biotechnology). All secondary antibodies were purchased either from Thermo Scientific (Molecular Probes) or Jackson ImmunoResearch Laboratory. During the last step of immunostaining, samples were counter stained with Hoechst33342 (1:1000, Cat#: 62249, RRID:AB_2651135, Thermo Scientific) solution in 1x PBS to visualize nuclei. Slides were mounted with Prolong gold anti-fade mounting medium (Cat#: P36930, RRID:SCR_015961, Thermo Scientific). All images were captured using a Nikon NiE-A1 plus or Nikon C2 confocal microscope, and analyzed using Image-J. For detailed inner ear histology protocols, please refer to our previous study (*Liu et al., 2010*). RNA in situ hybridization was performed according to protocols described previously (*Li et al., 2018*). The detailed sequence of the RNA probe for each gene was listed in *Supplementary file 3*.

## Quantification and statistical analysis of tdTomato+ SGNs in *Celf4-iCreER/+; Rosa26-CAG-loxp-stop-loxp-tdTomato /+* mice

To quantify the number of tdTomato+ SGNs in the distinct cochlear turns of *Celf4-iCreER/+; Rosa26-CAG-loxp-stop-loxp-tdTomato /+* mice of different ages, we used a cryosectioning approach (*Figure 6* and *Figure 6—figure supplement 2D–E''*). To precisely discriminate SGNs in basal, middle, and apical turns, we analyzed only those inner ear cryosection slices that included all three cochlear turns (*Figure 6B,E and H*, and *Figure 6—figure supplement 2D–E''*). For each slice obtained from the same mouse, we determined, per cochlear turn, the number of dual tdTomato+/ Tuj1+ (or Map2+ or Mafb+) SGNs, and then normalized these numbers against the total number of Tuj1+ (or Map2+ or Mafb+) SGNs to obtain an averaged percentage of tdTomato+ SGNs. These results are presented as means ± SEM (*Figure 6K–M*, and *Figure 6—figure supplement 2F*, n = 3 mouse numbers/age). Statistical analyses were performed using a one-way ANOVA, followed by a Student's *t* test with Bonferroni correction. GraphPad Prism 6.0 was used for all statistical analyses.

## Acknowledgements

We thank Dr. Qian Hu and the Optical Imaging Facility from the Institute of Neuroscience for support with the image analysis, Dr. Hui Yang (Principal Investigator from the Institute of Neuroscience) for sharing the zygote microinjection system to generate the knockin mice, Ms. Wenqin Ying (from the laboratory of Dr. Hui Yang at the Institute of Neuroscience) for helping us in transplanting zygotes into pseudopregnant female mice, and Dr. Virginia MS Rutten (HHMI-Janelia Research Campus, Ashburn, VA, USA) for assistance in editing the text.

## Additional information

### Funding

| Funder | Grant reference number | Author |
|---|---|---|
| Ministry of Science and Technology of the People's Republic of China | 2017YFA0103901 | Zhiyong Liu |
| Chinese Academy of Sciences | XDB32060100 | Zhiyong Liu |
| National Natural Science Foundation of China | 81771012 | Zhiyong Liu |
| Shanghai Municipal Education Commission | 2018SHZDZX05 | Zhiyong Liu |
| Boehringer Ingelheim | DE811138149 | Zhiyong Liu |

The funders had no role in study design, data collection and interpretation, or the decision to submit the work for publication.

### Author contributions
Chao Li, Xiang Li, Zhenghong Bi, Data curation, Formal analysis, Investigation, Writing - review and editing; Ken Sugino, Software, Formal analysis, Writing - original draft, Writing - review and editing; Guangqin Wang, Data curation, Formal analysis, Writing - review and editing; Tong Zhu, Data curation, Formal analysis; Zhiyong Liu, Conceptualization, Data curation, Formal analysis, Supervision, Funding acquisition, Validation, Investigation, Visualization, Methodology, Writing - original draft, Project administration, Writing - review and editing

### Author ORCIDs
Ken Sugino (iD) http://orcid.org/0000-0002-5795-0635
Zhiyong Liu (iD) https://orcid.org/0000-0002-9675-1233

### Ethics
Animal experimentation: All mice were bred and raised in SPF level animal rooms and animal procedures were performed according to guidelines (NA-032-2019) of the IACUC of Institute of Neuroscience (ION), Chinese Academy of Sciences.

### Decision letter and Author response
Decision letter https://doi.org/10.7554/eLife.50491.sa1
Author response https://doi.org/10.7554/eLife.50491.sa2

## Additional files
### Supplementary files
- Supplementary file 1. SGN genes with different dynamic patterns.
- Supplementary file 2. Genotyping and q-PCR primers.
- Supplementary file 3. Sequences of 8 RNA probes.
- Transparent reporting form

### Data availability
Sequencing data have been deposited in GEO under accession codes GSE132925.

The following dataset was generated:

| Author(s) | Year | Dataset title | Dataset URL | Database and Identifier |
|---|---|---|---|---|
| Li C, Li X, Bi Z, | 2019 | RNA-Seq of mouse inner ear SGNs, | https://www.ncbi.nlm. | NCBI Gene |

| Sugino K, Wang G, Zhu T, Liu Z | HCs and Glias | nih.gov/geo/query/acc. cgi?acc=GSE132925 | Expression Omnibus, GSE132925 |
| --- | --- | --- | --- |

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
