## [Decision Letter]

**Acceptance summary:**

Spiral ganglion neurons (SGNs) are the primary sensory neurons that mediate the sense of hearing in the mammalian inner ear. This study reports a new RNA-seq dataset for gene expression in developing and mature SGNs in the mouse. The authors have validated their approach by using CRISPR/Cas9 gene editing to generate two new knockin mouse strains for visualising and manipulating inner ear SGNs. Both the RNA-seq study and the new mouse strains will be valuable resources for those working on the mouse as a model for hearing and deafness. There was praise from the reviewers, whose comments include: 'Overall, there is clear value in such a transcriptomics analysis of SGNs…' 'The main strength of this project is the breadth of the dataset, which covers many time points as well as cell types.'

**Decision letter after peer review:**

Thank you for submitting your article "Comprehensive transcriptome analysis of cochlear spiral ganglion neurons at multiple ages" for consideration by *eLife*. Your article has been reviewed by three peer reviewers, and the evaluation has been overseen by a Reviewing Editor and Kathryn Cheah as the Senior Editor. The reviewers have opted to remain anonymous.

The reviewers have discussed the reviews with one another and the Reviewing Editor has drafted this decision to help you prepare a revised submission.

This manuscript sets out to provide a resource for the expression of genes at different developmental stages in spiral ganglion neurons of the mammalian cochlea. While all three reviewers agreed that the study had potential value for the field, they identified a number of issues that limit its impact at present. There is a need for some further experimental work and data analysis, together with some significant rewriting of the text. These revisions are required to make both the resource (the RNA-seq data) and the tools (the mice) more valuable to the community, and to point out advantages and applications over existing resources (e.g. online databases at NCBI and Jax).

The full reviews are appended below. The authors should aim to attend to all points, with particular emphasis on the following essential revisions:

1) Further validation of selected genes in both the constant and dynamic classes is required, using in situ hybridisation or immunohistochemistry.

2) Further analysis of both the Cre lines is required, in particular to characterise any expression in the CNS. Discussion should also be improved to point out any applications for which the new lines described here may be superior to existing lines.

3) Plans for distribution of the strains should be indicated.

Reviewer #1:

This manuscript presents a new RNA-seq dataset generated by collecting individual genetically labelled spiral ganglion neurons at 5 different ages. Cells were hand-picked to ensure purity and approximately 100 cells were collected per time period. Overall the quality of the data sets looks quite good. To validate the results, two genes, *Scrt2* and *Celf4*, were chosen to examine using a CRISPR-based to generate knock-in reporter lines. Again, the results appear good and suggest that the two lines will be useful for tracking/marking spiral ganglion neurons.

Major points:

1) It's not clear how many biological replicates were included for each time point.

2) Figure 8. It is not necessary to present some panels as pseudocolor because all colors in confocal images are pseudocolor.

3) Figure 6—figure supplement 1. Some higher magnification views of the tdTomato+ mesenchymal cells would be informative.

Reviewer #2:

Summary:

The authors identify genes specific to spiral ganglion neurons and determine changes in their expression over 5 timepoints by conducting bulk RNA-seq on manually picked and pooled SGNs, and at one time point for cochlear hair cells and glia. This strategy, while unable to distinguish Type I from Type II SGNs, permitted a much greater sequencing depth, and thus a more comprehensive transcriptome profile for what are presumably Type I cells than that obtained in other studies. This allowed identification of SGN-specific genes not found previously. This information is a valuable contribution to the inner ear field, but could be strengthened by additional computational analyses to illuminate whether there is enrichment for particular GO terms or biological processes that characterize the "constant" vs. "variable" gene groups and additional ISH validation. The authors targeted two of the newly identified genes for reporter or inducible Cre/reporter expression. Preliminary characterization of these mouse strains suggests that they may be useful to the field for marking SGNs or manipulating gene expression in SGNs, but some additional characterization of the lines (particularly the inducible Cre line) is needed.

Major points:

1) Judging from the Figure 2 legend, the 21 SGN-specific genes referred to paragraph one of subsection “Identifying new SGN genes with constant and time-varying expression pattern” are genes in the "constant" category, i.e. expressed in all SGN samples. This is not clear from the paragraph. Also, it is a bit disingenuous to state that those 21 genes were not reported in previous scRNA-seq studies because those studies were not aimed at discovering SGN-specific genes and the 21 genes may be expressed at levels too low for detection by scRNA-seq.

2) Given the relatively small number of genes on each list, it seems not unreasonable to conduct ISH (or immunostaining) validation of a few of the most highly expressed constant genes (e.g. *Elavl4, Tagln3* and *Esrrg*, or whatever the authors are most interested in) and a few of the most highly differentially expressed "variable" genes. Since the heat map in Figure 2B is normalized, one can't easily suggest which ones to test. A limited amount of ISH validation would increase confidence in the constant vs. variable categorization.

3) The Srct2-EGFP data in Figure 4 should be made supplemental because while it is interesting, it is not essential to the manuscript, and Supplementary Figure 3 should be merged with the current Figure 5, to show two time points at once in a main figure. Also, the authors need to determine whether *Scrt2* is expressed in all SGNs or only in Type I. This could be done with the HA-tag and SGN type-specific markers. The results would have implications for using this line to mark or purify cells, or when considering generation of an inducible Cre line.

4) To make the summary statement at the end of subsection “*Celf4* is dynamically expressed in both embryonic and adult SGNs”, varying expression over time needs to be shown for the *Celf4-iCreER* strain. This could be done by qRT-PCR or western botting. Also, the P30 data that are currently supplemental should be included in Figure 7.

5) The fate mapping experiment needs an additional control – *Celf4-iCreER/+;Rosa26-CAG-loxp-stop-loxp-tdTomato* (Ai9)/+ inner ears need to be checked for tdTomato in the absence of tamoxifen treatment to be sure that the iCre is not leaky. It would be very useful for the inner ear field to know more about the early expression of *Celf4*. When does it turn on in SGNs and at least for the E10.5 lineage induction, what proportion of SGNs are labeled?

Reviewer #3:

This manuscript reports a new dataset for gene expression in developing and mature spiral ganglion neurons (SGNs), the primary sensory neurons for the sense of hearing, as well as two new knock-in mouse strains for visualizing and manipulating inner ear neurons. Gene expression was assessed by bulk RNA-sequencing of pools of ~100 manually picked SGNs at five time points (E15.5, P1, P8, P14, P30) and then compared to gene expression in hair cells and glia to define SGN-enriched genes. The purity and quality of the samples was validated by qPCR for housekeeping, neuronal, glial, and hair cell genes (as well as some other categories), followed by characterization of reporter expression in two knock-in strains: *Scrt2*, which is expressed "constantly" in SGNs, and *Celf4*, which is expressed dynamically.

Overall, there is a clear value in such a comprehensive transcriptomic analysis of SGNs, but there a number of minor issues limit the likely impact of this particular data set. There is no doubt that having access to information about the cell type specificity and dynamics of expression of genes in SGNs is helpful for thinking about candidate genes or choosing markers. The main strength of this project is the breadth of the dataset, which covers many time points as well as cell types. With this resource, it will be relatively simple to see if genes of interest are likely to be expressed in other cell types and whether it is developmentally regulated. The data are generally high quality, in that there are 3 replicates for each sample and the qPCR validation suggests that the starting material was fairly pure, aside from hints of glial contamination in the later postnatal timepoints (Mbp expression in Figure 2—figure supplement 2).

That said, I am not entirely convinced that the data will have as broad an impact as intended. For instance, it might actually be easier for most people to simply check genes for expression by in situ hybridization, since in the end, only about 100 genes are identified as "SGN-specific." The mice also seem a bit less useful, but having another strain at your disposal is certainly nice, especially if there is a chance of avoiding pesky expression in the CNS. Finally, there are some misleading statements in the text that need to be corrected. More specific comments and requests for additional experiments/revisions follow.

1) Although the data are useful in principle, I was surprised to see how few genes ended up on the final lists of SGN-constant (21 genes) or dynamic (68 genes). The authors note that the list could be longer if the criteria were relaxed, but that raises the question of whether relaxing the criteria would diminish the value of the data, setting up a Catch-22. With so few genes, the value becomes lower and in the end, it might be easier for someone to just do some in situs for their gene of interest. I think it would also be worthwhile to define genes whose expression changes over time in SGNs even if they are also expressed in other cells. By screening out all genes that are expressed in HCs or glia, one loses many genes that are probably interesting and important for SGNs. Additionally, it looks like there may be some glial contamination from P8 on; more glial markers should be assessed at each time point.

2) There also needs to be more information regarding the contribution of Type II SGNs to the data. Presumably, the pool of SGNs contained mostly Type I SGNs but a few Type II SGNs. Since there are datasets highlighting Type II-specific genes, such as TH, the authors should show whether these genes are detected in the pools or whether the data are really dominated by Type I SGNs. This doesn't take away from the value; I would simply want to know how to interpret anything I see. This also highlights the fact that the currently available single cell RNA-seq data, when combined, likely cover the same 15,000 genes that are covered by the bulk RNA-seq, but without losing SGN subtype information. Thus, only the developmental stages provide truly new information.

3) With regards to the new mouse strains, these might be useful but they need further validation. Most importantly, there are already useful Cre lines for labeling and manipulating SGNs, such as *Neurog1-Cre, Shh-CreGFP, Bhlhe22-Cre* and *Mafb-CreERT2*, as well as a Mapt-GFP strain that can be used to specifically label SGNs at any stage. The authors state erroneously that *Mafb-CreERT2* only labels Type II SGNs in adults. While Mafb expression is indeed higher in Type IIs, it is also present in Type Is, as evidenced by the labeling shown in Shrestha et al., 2018, for instance. Additionally, the new strains here are also active in VGNs, unlike *Mafb-CreERT2*. The main issue with the lines out there is that they also label neurons in the CNS, which can be a big problem for tracing central projections or doing any kind of functional analysis. With that in mind, the *Scrt2* strain might be problematic, given the hints of expression in the hindbrain at E10.5. For both strains, the authors should also document expression patterns in the CNS, at least in the mature auditory brainstem.

4) The expression pattern in the *Scrt2-tdTomato* mice looks strange at E13.5. The authors should explain what all of the Map/tdTomato+ around the perimeter of the cochlea are (in Figure 2) as well as the section angle in Figure 4D, which I found hard to interpret.

5) The *Celf4* mouse strain could be useful if there is in fact little expression in the brainstem or elsewhere in the nervous system. The expression in the mesenchyme could raise problems in some cases, though (minor point). One question I had is whether there are conditions where all of the SGNs are labeled. It is surprising that only half of the neurons are labeled after 2 days of tamoxifen induction. It would be important to know if this is a random subset and whether there are conditions where in fact all neurons can be labeled. The inclusion of the HA-tag is a nice addition, but the results should really be validated by analyzing *Celf4* protein distribution using an independent method, such as antibody stain (minor – at least point out the caveat).

6) The presentation of the results stands to be improved in several ways. First, since this is a resource paper, more experimental information should be provided in the main text, including the number of SGNs per pool, the length of time from euthanasia to preservation in lysis buffer, and the criteria for SGN-constant vs dynamic. Second, there are some misstatements that gave me pause. Aside from the incorrect assertion that *Mafb-CreERT2* is Type II-specific in adults, the authors also intimate that the recent single cell RNA-seq datasets show changes in gene expression "underlying their different functions" (Abstract) but this is not accurate. No subtype-specific gene has been shown to endow subtype-specific functions, to my knowledge. Third, the authors cite the recent paper on *Pou4f1* (Sherrill et al., 2019) as evidence that *Pou4f1* regulates "IHC presynaptic calcium signaling". This is misleading: that paper showed changes in presynaptic calcium signaling in the IHCs of *Pou4f1* CKO mice, but since *Pou4f1* is expressed in the SGNs, the origins of this phenotype must be secondary. Fourth, it seems inappropriate to discuss results from an unpublished study (subsection “Applications of *Scrt2-tdTomato/+* knock-in mouse model”) as evidence that a new mouse strain is useful, as the reader can't independently assess the validity of the statements. Fifth, the authors mention several genes that were "not identified" by single cell RNA-seq, but then immediately cite those studies as evidence that the gene were expressed, for instance in both Type I and Type II SGNs. I think the authors mean that the genes were not emphasized as being subtype-specific, which is very different from saying they were not identified at all. Sixth, the authors missed the opportunity to emphasize an advantage of this new dataset over previous data, in that old microarray comparisons of SGNs and VGNs at multiple time points were almost certainly affected by the presence of glia in the samples (that is, if the SGN samples are indeed as pure as suggested – see point 1).

[Editors' note: further revisions were suggested prior to acceptance, as described below.]

Thank you for resubmitting your work entitled "Comprehensive transcriptome analysis of cochlear spiral ganglion neurons at multiple ages" for further consideration by *eLife*. Your revised article has been evaluated by Kathryn Cheah (Senior Editor) and a Reviewing Editor.

The manuscript has been improved but there are some remaining issues that need to be addressed before acceptance. After discussion between the reviewers and reviewing editor, it was agreed that essential changes are:

1) Please re-photograph the in situ hybridisation images in Figure 2—figure supplement 3 to improve the focus and illumination.

2) Please move Figure 2—figure supplement 4 to the main figures.

3) Please move Figure 3 (targeting strategy) to supplemental.

4). If possible, please show whole-mounts of the tdTomato staining. If this is not possible, please adjust the text to include discussion of the limitations of the Gata3 staining.

5) Please adjust the writing and nomenclature as suggested.

Detailed suggestions for all these points can be found in the comments from reviewers 2 and 3 below.

Reviewer #1:

The authors have addressed all of my concerns.

Reviewer #2:

I am more or less satisfied with the responses to the reviews. The authors answered each point and have improved the study's utility to the inner ear field. It would still be desirable to know more about Cre activity from the new lines in the auditory/vestibular centers in the brain, but I understand that this sort of detailed characterization will take future time and effort from the authors as well as the community of those who want to use the lines.

I remain somewhat disappointed by the writing, which detracts from the impact. Nevertheless, given that this is a resources paper, it presents very useful data to the inner ear field and describes two new SGN Cre lines that will be made available through JAX. These complement available lines and are likely to be useful.

Reviewer #3:

The authors have done an excellent job of addressing my concerns. They have added interesting new data that a) confirm expression of 8 genes by in situ hybridization; b) identify groups of developmentally regulated genes in SGNs; c) show expression of both new mouse lines in Type I and Type II SGNs; and d) show that the two mouse lines also drive expression elsewhere in the nervous system, with an expanded discussion of advantages in the Discussion. I particularly appreciate the addition of Figure 2—figure supplement 4, which highlights many other groups of genes uncovered by this analysis. As a result, the value of the data and of the new mouse lines is significantly enhanced. I have only a few criticisms.

1) The authors often use the term "SGN-specific" when in fact, at least some of the genes are also expressed in vestibular ganglion neurons. To avoid confusion, the authors should explain up front that some genes may also be expressed in VGNs. Also, I think it is confusing to use the term "VNs". VGNs parallels the SGN nomenclature more accurately.

2) Subsection “Isolated and purified SGNs are highly enriched in neuronal genes and depleted in HC and glial genes”: we "totally used" should be "used a total of".

3) Subsection “Identifying SGN specific genes with constant and dynamic expression pattern”: "at all SGNs" isn't proper English. Also, I don't think these data can resolve whether expression is in all SGNs. Perhaps the authors meant "at all timepoints examined"?

4) Figure 2—figure supplement 3 is an excellent addition. However, the images are poor quality. New images should be taken.

5) Though the data in Figure 8 look convincing, Gata3 is not a reliable Type II SGN marker. If possible, the authors should show wholemounts with tdTomato-labeled processes. This will definitively indicate whether both Type I and Type II SGNs are labeled, as they can be reliably distinguished by their projection patterns.

6) Figure 2—figure supplement 4 is very interesting and increases the value of the study. I feel this figure should not be supplemental. The diagrams of the modified gene loci plus Southern blots and PCRs, on the other hand, could be supplemental. My suggestion is to just show the modified locus as a panel in the first figure showing expression from the relevant mouse line, and to move the targeting strategy and Southern blot/PCR validation to supplemental.

---

## [Author Response]

Reviewer #1:This manuscript presents a new RNA-seq dataset generated by collecting individual genetically labelled spiral ganglion neurons at 5 different ages. Cells were hand-picked to ensure purity and approximately 100 cells were collected per time period. Overall the quality of the data sets looks quite good. To validate the results, two genes, Scrt2 and Celf4, were chosen to examine using a CRISPR-based to generate knock-in reporter lines. Again, the results appear good and suggest that the two lines will be useful for tracking/marking spiral ganglion neurons.Major points:1) It's not clear how many biological replicates were included for each time point.

Thanks for pointing out this unclear part. At each time point, three replicates were performed. Please refer to paragraph two of subsection “Isolated and purified SGNs are highly enriched in neuronal genes and depleted in HC and glial genes”.

2) Figure 8. It is not necessary to present some panels as pseudocolor because all colors in confocal images are pseudocolor.

We agree with this comment and have deleted the word “pseudocolor”.

3) Figure 6—figure supplement 1. Some higher magnification views of the tdTomato+ mesenchymal cells would be informative.

We agree with this suggestion and have added 60x confocal image of mesenchymal cells, please refer to the Figure 6—figure supplement 1.

Reviewer #2:Summary:The authors identify genes specific to spiral ganglion neurons and determine changes in their expression over 5 timepoints by conducting bulk RNA-seq on manually picked and pooled SGNs, and at one time point for cochlear hair cells and glia. This strategy, while unable to distinguish Type I from Type II SGNs, permitted a much greater sequencing depth, and thus a more comprehensive transcriptome profile for what are presumably Type I cells than that obtained in other studies. This allowed identification of SGN-specific genes not found previously. This information is a valuable contribution to the inner ear field, but could be strengthened by additional computational analyses to illuminate whether there is enrichment for particular GO terms or biological processes that characterize the "constant" vs. "variable" gene groups and additional ISH validation. The authors targeted two of the newly identified genes for reporter or inducible Cre/reporter expression. Preliminary characterization of these mouse strains suggests that they may be useful to the field for marking SGNs or manipulating gene expression in SGNs, but some additional characterization of the lines (particularly the inducible Cre line) is needed.

We appreciated the encouraging comments of our studies as well as suggestion above. In the revised manuscript, we have performed additional analyses and experiments accordingly: 1) GO analysis was described in subsection “Classifying dynamics of genes expressed in SGNs” and Figure 2—figure supplement 4K-M; 2) we further selected 8 genes and validated their specific expression in SGNs by RNA in situ hybridization, please refer to subsection “Identifying SGN specific genes with constant and dynamic expression pattern” and Figure 2—figure supplement 3; 3) We additional analyzed *Celf4-iCreER* line by administering tamoxifen at E10.5 (Subsection “Fate mapping analysis by using *Celf4 iCreER*/+ mouse strain” and Figure 7—figure supplement 2) and expression of *Scrt2* and *Celf4* at adult brains (subsection “*Scrt2* and *Celf4* are expressed in both type I and type II SGNs at adult ages” and Figure 8—figure supplement 1).

Major points:1) Judging from the Figure 2 legend, the 21 SGN-specific genes referred to paragraph one of subsection “Identifying new SGN genes with constant and time-varying expression pattern” are genes in the "constant" category, i.e. expressed in all SGN samples. This is not clear from the paragraph. Also, it is a bit disingenuous to state that those 21 genes were not reported in previous scRNA-seq studies because those studies were not aimed at discovering SGN-specific genes and the 21 genes may be expressed at levels too low for detection by scRNA-seq.

Thanks for pointing this out and we have re-written this paragraph. Moreover, throughout the manuscript, we deleted the sentences “….those 21 genes were not reported in previous scRNA-seq studies…”. Instead,we stated that previously unknown SGN specific genes were identified.

2) Given the relatively small number of genes on each list, it seems not unreasonable to conduct ISH (or immunostaining) validation of a few of the most highly expressed constant genes (e.g. Elavl4, Tagln3 and Esrrg, or whatever the authors are most interested in) and a few of the most highly differentially expressed "variable" genes. Since the heat map in Figure 2B is normalized, one can't easily suggest which ones to test. A limited amount of ISH validation would increase confidence in the constant vs. variable categorization.

Thanks for this suggestion. We have selected 8 genes (4 are SGN specific showing constant pattern, 4 are SGN specific showing dynamic pattern) to perform RNA in situ hybridization. All of the 8 genes were specifically expressed in SGN region. Please refer to subsection “Identifying SGN specific genes with constant and dynamic expression pattern” and Figure 2—figure supplement 3.

3) The Srct2-EGFP data in Figure 4 should be made supplemental because while it is interesting, it is not essential to the manuscript, and Supplementary Figure 3 should be merged with the current Figure 5, to show two time points at once in a main figure. Also, the authors need to determine whether Scrt2 is expressed in all SGNs or only in Type I. This could be done with the HA-tag and SGN type-specific markers. The results would have implications for using this line to mark or purify cells, or when considering generation of an inducible Cre line.

Thanks for these suggestions above. We have re-organized those figures and please refer to the Figure 4 and Figure 4—figure supplement 1. Furthermore, we confirmed that *Scrt2* and *Celf4* are expressed in both Type I and II SGNs at adult ages.

4) To make the summary statement at the end of subsection “Celf4 is dynamically expressed in both embryonic and adult SGNs”, varying expression over time needs to be shown for the Celf4-iCreER strain. This could be done by qRT-PCR or western botting. Also, the P30 data that are currently supplemental should be included in Figure 7.

Thanks for these suggestions and we performed qPCR of *Celf4*. Please refer to subsection “*Celf4* is dynamically expressed in both embryonic and adult SGNs” and Figure 2D. In addition, P30 data (together with E15.5 and P1) of *Celf4-iCreER* strain were presented in the Figure 6.

5) The fate mapping experiment needs an additional control – Celf4-iCreER/+;Rosa26-CAG-loxp-stop-loxp-tdTomato (Ai9)/+ inner ears need to be checked for tdTomato in the absence of tamoxifen treatment to be sure that the iCre is not leaky. It would be very useful for the inner ear field to know more about the early expression of Celf4. When does it turn on in SGNs and at least for the E10.5 lineage induction, what proportion of SGNs are labeled?

Thanks for these suggestions. We have performed the additional control (no tamoxifen) in *Celf4-iCreER/+; Rosa26-CAG-loxp-stop-loxp-tdTomato/+* mice. No tdTomato+ SGNs were observed at P1. This confirmed that *Celf4-iCreER* was not leaky. In contrast, when tamoxifen was administered to *Celf4-iCreER/+; Rosa26-CAG-loxp-stop-loxp-tdTomato/+* mice at E10.5, numerous tdTomato+ SGNs were observed at E18.5. Please refer to subsection “Fate mapping analysis by using *Celf4 iCreER/+* mouse strain” and Figure 7—figure supplement 2.

Reviewer #3:[…]1) Although the data are useful in principle, I was surprised to see how few genes ended up on the final lists of SGN-constant (21 genes) or dynamic (68 genes). The authors note that the list could be longer if the criteria were relaxed, but that raises the question of whether relaxing the criteria would diminish the value of the data, setting up a Catch-22. With so few genes, the value becomes lower and in the end, it might be easier for someone to just do some in situs for their gene of interest. I think it would also be worthwhile to define genes whose expression changes over time in SGNs even if they are also expressed in other cells. By screening out all genes that are expressed in HCs or glia, one loses many genes that are probably interesting and important for SGNs. Additionally, it looks like there may be some glial contamination from P8 on; more glial markers should be assessed at each time point.

We appreciated this suggestion. We have performed additional computational analysis of SGN genes without considering whether they are expressed in HCs and glias. GO enrichment analyses were performed, too. Please refer to subsection “Classifying dynamics of genes expressed in SGNs” and Figure 2—figure supplement 4K-M. In terms of the glia contamination, we totally checked four known glial markers *Sox2* (also in vestibular HCs), Sox10, Foxd3 and Plp1 (Figure 2—figure supplement 2). Currently, much less glial markers are known than those of SGNs or HCs. Future RNA-Seq of glial cells at multiple ages will reveal more glia specific genes.

2) There also needs to be more information regarding the contribution of Type II SGNs to the data. Presumably, the pool of SGNs contained mostly Type I SGNs but a few Type II SGNs. Since there are datasets highlighting Type II-specific genes, such as TH, the authors should show whether these genes are detected in the pools or whether the data are really dominated by Type I SGNs. This doesn't take away from the value; I would simply want to know how to interpret anything I see. This also highlights the fact that the currently available single cell RNA-seq data, when combined, likely cover the same 15,000 genes that are covered by the bulk RNA-seq, but without losing SGN subtype information. Thus, only the developmental stages provide truly new information.

We appreciated this suggestion. We checked several Type II markers: Mafb, Gata3, Th, and *Calca/Cgrpα* expression in SGN_P30 replicates. Their averaged TPM was 21, 8, 1.9 and 1.8, respectively. Due to ~5% distribution of Type II SGNs at adult ages, we on average may only picked ~5 type II SGNs in each replicate (with each replicate ~100 cells). In this situation (Type II SGN genes are diluted a lot), Type II SGNs are very likely to be included.

3) With regards to the new mouse strains, these might be useful but they need further validation. Most importantly, there are already useful Cre lines for labeling and manipulating SGNs, such as Neurog1-Cre, Shh-CreGFP, Bhlhe22-Cre and Mafb-CreERT2, as well as a Mapt-GFP strain that can be used to specifically label SGNs at any stage. The authors state erroneously that Mafb-CreERT2 only labels Type II SGNs in adults. While Mafb expression is indeed higher in Type IIs, it is also present in Type Is, as evidenced by the labeling shown in Shrestha et al., 2018, for instance. Additionally, the new strains here are also active in VGNs, unlike Mafb-CreERT2. The main issue with the lines out there is that they also label neurons in the CNS, which can be a big problem for tracing central projections or doing any kind of functional analysis. With that in mind, the Scrt2 strain might be problematic, given the hints of expression in the hindbrain at E10.5. For both strains, the authors should also document expression patterns in the CNS, at least in the mature auditory brainstem.

We have further characterized the expression of *Scrt2* and *Celf4* in adult central nervous system. Please refer to subsection “*Scrt2* and *Celf4* are expressed in adult brain tissues” and the Figure 8—figure supplement 1. We also discussed the differences among our two lines and available ones. We will characterize expression of *Scrt2* and *Celf4* in adult auditory brainstem in future studies. Our laboratory did not have any previous experience about adult auditory brainstem and it will take a while for us to be familiar with the histology and how to precisely handle auditory brainstem tissues out.

4) The expression pattern in the Scrt2-tdTomato mice looks strange at E13.5. The authors should explain what all of the Map/tdTomato+ around the perimeter of the cochlea are (in Figure 2) as well as the section angle in Figure 4D, which I found hard to interpret.

Sorry for confusing.Those are the autofluorescence of the blood cells in vessels in cochlea, as labeled by asterisks in the Figure 4 and Figure 4—figure supplement 1. Please also refer to text in the corresponding figure legends.

5) The Celf4 mouse strain could be useful if there is in fact little expression in the brainstem or elsewhere in the nervous system. The expression in the mesenchyme could raise problems in some cases, though (minor point). One question I had is whether there are conditions where all of the SGNs are labeled. It is surprising that only half of the neurons are labeled after 2 days of tamoxifen induction. It would be important to know if this is a random subset and whether there are conditions where in fact all neurons can be labeled. The inclusion of the HA-tag is a nice addition, but the results should really be validated by analyzing Celf4 protein distribution using an independent method, such as antibody stain (minor – at least point out the caveat).

Because *Celf4-iCreER* is an inducible Cre, it will be difficult to label all SGNs. But if more than two tamoxifen administering was given, more SGNs should be labeled. In addition, according to RNA in situ data (Figure 2—figure supplement 3), *Celf4* expression level is relatively lower than other SGN specific genes. Currently, there is no good *Celf4* antibody suitable for immunostaining, and that is why we added HA tag to make future studies more convenient.

6) The presentation of the results stands to be improved in several ways. First, since this is a resource paper, more experimental information should be provided in the main text, including the number of SGNs per pool, the length of time from euthanasia to preservation in lysis buffer.

We appreciated this suggestion. We added relevant information above.

The authors also intimate that the recent single cell RNA-seq datasets show changes in gene expression "underlying their different functions" (Abstract) but this is not accurate. No subtype-specific gene has been shown to endow subtype-specific functions, to my knowledge.

Thanks for pointing this out. We have edited this sentence.

Third, the authors cite the recent paper on Pou4f1 (Sherrill et al., 2019) as evidence that Pou4f1 regulates "IHC presynaptic calcium signaling". This is misleading: that paper showed changes in presynaptic calcium signaling in the IHCs of Pou4f1 CKO mice, but since Pou4f1 is expressed in the SGNs, the origins of this phenotype must be secondary.

Thanks for pointing this out. We have deleted discussion of *Pou4f1* in SGNs.

Fourth, it seems inappropriate to discuss results from an unpublished study (subsection “Applications of Scrt2-tdTomato/+ knock-in mouse model”) as evidence that a new mouse strain is useful, as the reader can't independently assess the validity of the statements.

We have deleted information about our ongoing studies of SGN induction. This is also raised by reviewer 2.

Fifth, the authors mention several genes that were "not identified" by single cell RNA-seq, but then immediately cite those studies as evidence that the gene were expressed, for instance in both Type I and Type II SGNs. I think the authors mean that the genes were not emphasized as being subtype-specific, which is very different from saying they were not identified at all.

Sorry for that. Reviewer 2 also pointed this out. Thank you very much. Throughout the revised manuscript, we deleted the sentences saying like “….those SGN specific genes were not reported in previous scRNA-seq studies…”.Instead,we stated that previously unknown SGN specific genes were identified.

Sixth, the authors missed the opportunity to emphasize an advantage of this new dataset over previous data, in that old microarray comparisons of SGNs and VGNs at multiple time points were almost certainly affected by the presence of glia in the samples (that is, if the SGN samples are indeed as pure as suggested – see point 1).

Thanks for this suggestion. In addition, GO analysis of both our dataset and microarray data showed the enrichment of immune-related genes, which in turn can support the purity of our SGNs.

[Editors' note: further revisions were suggested prior to acceptance, as described below.]

Reviewer #2:I am more or less satisfied with the responses to the reviews. The authors answered each point and have improved the study's utility to the inner ear field. It would still be desirable to know more about Cre activity from the new lines in the auditory/vestibular centers in the brain, but I understand that this sort of detailed characterization will take future time and effort from the authors as well as the community of those who want to use the lines.I remain somewhat disappointed by the writing, which detracts from the impact. Nevertheless, given that this is a resources paper, it presents very useful data to the inner ear field and describes two new SGN Cre lines that will be made available through JAX. These complement available lines and are likely to be useful.

We appreciated the encouraging comments of our revision. In near future, we will start to learn the histology of auditory/vestibular centers in the brain and we will report detailed patterns of *Scrt2* and *Celf4* in auditory/vestibular centers in our next manuscript.

Reviewer #3:[…]1) The authors often use the term "SGN-specific" when in fact, at least some of the genes are also expressed in vestibular ganglion neurons. To avoid confusion, the authors should explain up front that some genes may also be expressed in VGNs. Also, I think it is confusing to use the term "VNs". VGNs parallels the SGN nomenclature more accurately.

Thanks for pointing this out. We have added the sentence to clearly say that although those genes were “SGN-specific” within cochlea organ, they were also expressed in vestibular neurons and neurons in central nervous system. In addition, we have changed VNs to VGNs throughout the manuscript.

2) Subsection “Isolated and purified SGNs are highly enriched in neuronal genes and depleted in HC and glial genes”: we "totally used" should be "used a total of".

“totally used” was changed to "used a total of".

3) Subsection “Identifying SGN specific genes with constant and dynamic expression pattern”: "at all SGNs" isn't proper English. Also, I don't think these data can resolve whether expression is in all SGNs. Perhaps the authors meant "at all timepoints examined"?

Thanks for pointing this out. We have changed “at all SGNs” to “in SGNs at all timepoints examined”.

4) Figure 2—figure supplement 3 is an excellent addition. However, the images are poor quality. New images should be taken.

We have re-taken the images at 20x magnification, please see the updated Figure 2—figure supplement 3.

5) Though the data in Figure 8 look convincing, Gata3 is not a reliable Type II SGN marker. If possible, the authors should show wholemounts with tdTomato-labeled processes. This will definitively indicate whether both Type I and Type II SGNs are labeled, as they can be reliably distinguished by their projection patterns.

Thanks for pointing this out. We have additionally performed whole mount double staining of Gata3 and tdTomato, and please see the new Figure 7. We agreed that, besides in SGNs, Gata3 is also expressed in many other cells in cochlea such as hair cells and supporting cells (Figure. 7C). We did not present whole mount staining of Gata3 and tdTomato in *Scrt2-P2A-tdTomato/+* samples at P30, because the tdTomato is not bright enough to visualize neuronal fiber terminals.

In addition, we pointed out that possibility that some of the tdTomato+ fibers might be cochlear efferent fibers (Figure 7D-D’), if *Celf4* is also expressed in brainstem. Future detailed work is needed to clarify it.

6) Figure 2—figure supplement 4 is very interesting and increases the value of the study. I feel this figure should not be supplemental. The diagrams of the modified gene loci plus Southern blots and PCRs, on the other hand, could be supplemental. My suggestion is to just show the modified locus as a panel in the first figure showing expression from the relevant mouse line, and to move the targeting strategy and Southern blot/PCR validation to supplemental.

Thanks for these suggestions and we have accepted them. The previous Figure 2—figure supplement 4 were Figure 3 in the revised manuscript. In addition, we changed previous main figures of Southern blot and PCR (for both *Scrt2* and *Celf4* strains) into supplemental figures.